# When Lifelong Novelty Fails: Coordination Breakdown in Decentralised MARL

**Ting Zhu[1], Yue Jin[2], Giovanni Montana[1,3,4]**

[1]*Department of Statistics, University of Warwick, Coventry, UK*
[2]*Faculty of Arts, Science & Technology, University of Northampton, Northampton, UK*
[3]*Warwick Manufacturing Group, University of Warwick, Coventry, UK*
[4]*Alan Turing Institute, London, UK*

*ting.zhu@warwick.ac.uk*
*esther.jin@northampton.ac.uk*
*g.montana@warwick.ac.uk*

**Reviewed on OpenReview:** *https://openreview.net/forum?id=xOPjPFTuvy*

## Abstract

Lifelong novelty bonuses are a cornerstone of exploration in reinforcement learning, but we identify a critical failure mode when they are applied to decentralised multi-agent co-ordination tasks: *coordination de-synchronisation*. In sequential coordination tasks with multiple joint coordination checkpoints (states that all agents must occupy simultaneously), agents searching for later checkpoints must repeatedly traverse earlier ones. Under life-long novelty, this repeated traversal gradually depletes intrinsic motivation to revisit these critical locations and can destabilise coordination. Within a stylised analytical framework, we derive lower bounds showing that the *guaranteed* success probability under a lifelong novelty scheme can shrink polynomially with a problem-dependent geometric *revisit pressure* and the number of agents, whereas episodic bonuses, which reset at the start of each episode, provide a time-uniform lower bound on the probability of reaching a given check-point. We further prove that a hybrid scheme, which multiplicatively combines episodic and lifelong bonuses, inherits a finite-window lower bound on re-coordination probability at known checkpoints, under an episodic revisit-cap assumption, while preserving a drive to discover previously unseen states. We validate the qualitative predictions of this framework in GridWorld, Overcooked, and StarCraft II, where hybrid bonuses yield substantially more reliable coordination than lifelong-only exploration in environments with multiple sequen-tial checkpoints or narrow geometric bottlenecks, such as corridors that force agents to pass through the same cells many times. Together, these results provide a theoretical and em-pirical account of when different intrinsic motivation schemes are effective in decentralised multi-agent coordination.

## 1 Introduction

Cooperative multi-agent reinforcement learning (MARL) presents unique challenges beyond single-agent settings, particularly in decentralised scenarios where agents must learn coordination strategies based solely on local observations without global awareness. Despite the progress made by recent fully decentralised algorithms (De Witt et al., 2020; Jiang & Lu, 2022; Zhu et al., 2024), a significant limitation arises in environments with sparse rewards. Exploration becomes especially critical in sparse-reward cooperative tasks, where agents must synchronise their actions at specific locations and times to make progress, yet only receive exceedingly rare external reward signals and cannot infer other agents' states from a global view.

A common approach to exploration in reinforcement learning is through intrinsic motivation via novelty-based bonuses: additional rewards based on how rarely states have been visited (Bellemare et al., 2016; Pathak et al., 2017; Burda et al., 2019). These bonuses encourage agents to explore unfamiliar regions of the state space. Most existing methods employ *lifelong novelty*, which tracks state visits over the entire course of training and permanently reduces the bonus for a state once it becomes familiar. Whilst effective in many single-agent scenarios, this permanent devaluation creates problems when agents need to revisit important states multiple times (Henaff et al., 2023). An alternative approach, *episodic novelty*, resets visit counts at the beginning of each episode (Stanton & Clune, 2018; Raileanu & Rocktäschel, 2020; Henaff et al., 2022; Zhang et al., 2021), maintaining exploration incentives for states that remain important throughout training.

Extending these methods to decentralised multi-agent settings introduces additional complexity (Hao et al., 2023). Under decentralised execution, each agent computes novelty from its own local state or observation. The issue is not reward locality by itself, since environment rewards may also be observed locally in MARL. Rather, the specific difficulty for novelty bonuses is that local novelty can be misaligned with the joint-state novelty that matters for coordination: an agent may regard its own state as familiar or novel even when the corresponding joint configuration is critical for team progress. This gap between local and global perspectives can misalign exploration incentives across the team. Moreover, in tasks requiring sequential coordination—where agents must synchronise at multiple *coordination checkpoints*, defined as joint states that all agents must occupy simultaneously—the interaction between novelty-based exploration and coordination requirements becomes critical. Yet there is little theoretical understanding of how different novelty schemes affect multi-agent coordination in such settings.

In this work, we identify a fundamental failure mode of lifelong novelty in decentralised multi-agent exploration: *coordination de-synchronisation*. This occurs in tasks requiring agents to coordinate at multiple sequential checkpoints. As agents explore to find later checkpoints, they must repeatedly traverse earlier ones. Under lifelong novelty, this repeated traversal gradually exhausts their intrinsic motivation to revisit these critical locations. The result is a coordination "death spiral": the very exploration needed to discover advanced objectives undermines the team's ability to maintain coordination at foundational checkpoints.

We emphasise that the multiplicative combination of episodic and lifelong novelty is not itself our algorithmic contribution. Related single-agent and multi-agent work has already studied such hybrid novelty signals. Our contribution is instead to identify and analyse a decentralised MARL failure mode—coordination de-synchronisation—and to show how the temporal scope of novelty interacts with sequential coordination structure. Our key insight is that coordination de-synchronisation depends on two independent factors: the task's *coordination complexity* (the number of sequential checkpoints agents must reach together) and the environment's *geometric structure*, which determines how often agents must revisit earlier checkpoints whilst searching for later ones. Within a stylised analytical framework, we show that the guaranteed success probability under a lifelong novelty scheme can shrink polynomially with a problem-dependent geometric *revisit pressure* and the number of agents, and that our sufficient conditions ensure finite expected hitting time only in a restricted regime. In contrast, we show that a hybrid approach—multiplicatively combining episodic and lifelong bonuses—provides a robust finite-window mechanism: the episodic component suppresses excessive within-episode cycling and thereby slows depletion of task-critical revisit states, while the lifelong component preserves the exploratory drive needed to discover new regions. Thus, the episodic term is preventive and trajectory-shaping rather than an asymptotic mechanism for restoring a lifelong bonus that has already vanished.

We validate the qualitative predictions of this framework across GridWorld, Overcooked, and StarCraft II environments, demonstrating that hybrid approaches achieve substantially faster and more reliable coordination than lifelong-only baselines in challenging sequential scenarios with high revisit pressure. Our experiments reveal sharp degradation patterns for lifelong novelty that are consistent with our theoretical sufficient conditions, and stable performance for hybrid bonuses. Taken together, this provides a theoretical and empirical account of when and why different intrinsic motivation schemes succeed or fail in decentralised multi-agent coordination, offering principled guidance for exploration in cooperative settings.

## 2    Related work

**Intrinsic motivation in single-agent RL.**    Intrinsic motivation through novelty-based exploration has been extensively studied in single-agent reinforcement learning. Most work focuses on *lifelong novelty*, using inverse visit counts in tabular settings (Bellemare et al., 2016; Ostrovski et al., 2017; Machado et al., 2020) or alternative approaches including prediction error (Pathak et al., 2017), state marginal matching (Lee et al., 2019), and Bayesian surprise (Mazzaglia et al., 2022). These methods permanently devalue states once they become familiar.

Stanton & Clune (2018) distinguish between inter-episode (lifelong) and intra-episode (episodic) novelty. Badia et al. (2020) introduced Never Give Up (NGU), which multiplicatively combines episodic and lifelong novelty in single-agent reinforcement learning to support directed exploration in hard-exploration domains.

More recent work has focused on episodic-level exploration, where agents reset novelty estimates each episode (Henaff et al., 2022; Savinov et al., 2019; Jiang et al., 2025). However, these single-agent studies do not address coordination challenges in multi-agent settings, nor do they analyse when different temporal scopes become crucial for task success. Our work extends these concepts to decentralised MARL and proposes a stylised theoretical characterisation of when episodic bonuses can become essential, in terms of coordination complexity and environment structure.

**Intrinsic motivation in multi-agent RL.**    Existing multi-agent intrinsic motivation work primarily operates within the Centralised Training with Decentralised Execution (CTDE) framework, where global information is available during training. Iqbal & Sha (2019) propose coordinated exploration bonuses that take other agents' perspectives into account. Wang* et al. (2020) use mutual information between agents' transitions, whilst Zheng et al. (2021) employ prediction errors of individual Q-functions. Liu et al. (2023) address lazy agents through causal effect maximisation.

These CTDE approaches differ fundamentally from our fully decentralised setting, where agents rely solely on local observations. Moreover, there is little theoretical analysis of when different exploration strategies succeed or fail as a function of task structure. Jiang et al. (2024) introduce MACE as a method specifically designed for decentralised settings, where agents share local novelty values. However, it focuses on approximating global novelty rather than addressing the coordination challenges we identify. Our work instead offers a theoretical framework that highlights when episodic components can be important for decentralised coordination.

**Episodic bonuses in MARL.**    Toquebiau et al. (2024) combine lifelong and episodic bonuses in MARL, using NovelD (Zhang et al., 2021) and E3B (Henaff et al., 2022) within CTDE. However, they compute novelty using joint observations, giving all agents identical bonuses. Our decentralised approach requires agents to compute individual novelty from local observations. Hernandez et al. (2025) study NGU-style exploration in multi-agent reinforcement learning, including implementation choices such as replay-buffer sharing. Our work is complementary: rather than proposing NGU as a new mechanism, we analyse why the interaction between lifelong and episodic novelty matters for decentralised sequential coordination. Their empirical study does not analyse when combinations of episodic and lifelong bonuses are particularly beneficial, whereas our framework focuses on conditions under which combining episodic and lifelong bonuses improves coordination in decentralised settings.

**Communication in decentralised MARL.**    Research on decentralised communication has focused on establishing common grounding between agents. Lin et al. (2021) propose AEComm, where agents autoencode observations for communication using shared environment structure, whilst Lo et al. (2023) introduce contrastive learning to align messages from agents perceiving the same global state. Our theoretical analysis considers the fundamental case where agents operate independently without communication, and illustrates how coordination complexity and revisit pressure shape the effectiveness of different intrinsic motivation schemes. While our experiments extend this to include novelty sharing between agents, the core insight—that episodic bonuses provide coordination robustness whilst lifelong bonuses are vulnerable in sequential tasks—appears to hold regardless of whether agents share novelty information. This suggests that the temporal scope of novelty assessment may be more fundamental than the specific communication mechanism.

## 3 Theoretical framework

### 3.1 Problem Formulation

We consider an $N$-agent cooperative Markov decision process $\mathcal{M} = (\mathcal{S}, \mathcal{A}, P, R, \gamma)$, where $\mathcal{S}$ is the joint state space, $\mathcal{A} = \mathcal{A}_1 \times \cdots \times \mathcal{A}_N$ is the joint action space, $P(s' \mid s, a)$ is the transition kernel, $R(s, a)$ is a shared team reward, and episodes have horizon $H$. We study decentralised execution: agent $i$ selects actions using only its local state $s_i$, according to a policy $\pi_i(a_i \mid s_i)$, without observing the full joint state $s$ or the other agents' local states $s_{-i}$.

From the perspective of agent $i$, the effective dynamics depend on the policies of the other agents,

$$P_i(s' \mid s, a_i) = \sum_{a_{-i}} P(s' \mid s, (a_i, a_{-i})) \pi_{-i}(a_{-i} \mid s_{-i}),$$

where $\pi_{-i}$ denotes the product policy of all agents except $i$. As these policies change during learning, each agent experiences non-stationary effective dynamics. Our analysis focuses on the common subclass in which each local observation uniquely determines the corresponding local state $s_i$, such as an agent's grid position and local door status. The setting is still globally partially observable because agents do not observe $s_{-i}$.

We restrict attention to fully cooperative tasks because the analysis relies on aligned incentives: all agents optimise the same sparse team reward, and intrinsic rewards are intended to bias the joint policy toward states useful for team progress. In mixed or competitive games, a state that is novel for one agent may be strategically harmful, exploitable, or irrelevant to another agent. In such settings, the monotone relationship between intrinsic bonus and useful joint visitation assumed below need not hold, and different game-theoretic assumptions would be required.

### 3.2 Coordination complexity and success metrics

We study cooperative tasks that require agents to visit an ordered sequence of *coordination checkpoints*, $S^* = \{s^*_{(1)}, \ldots, s^*_{(L)}\}$. Each checkpoint is a joint state $s^*_{(\ell)} = (s^*_{1,\ell}, \ldots, s^*_{N,\ell})$ where all agents must be simultaneously present. For instance, a simple task might require two agents to stand on separate pressure plates to open a door. This joint requirement—Agent 1 at position A *and* Agent 2 at position B—constitutes the first checkpoint, $s^*_{(1)}$. The notation $s^*_{i,\ell}$ indexes the local component of the joint checkpoint for agent $i$; it does not imply that behavioural roles are manually assigned in advance. In our experiments, agent identities are fixed, but roles such as "activating a switch", "passing through a door", or "performing a handoff" emerge through learning. For homogeneous agents, several role assignments may be valid. In that case, a checkpoint can equivalently be represented as a set of valid joint states, including permutations of the agents' physical roles. Our analysis applies to any feasible assignment, and allowing multiple feasible assignments only increases the probability of reaching the checkpoint. For heterogeneous agents, feasible local state and action spaces are agent-specific; this changes only the behavioural and task-dependent constants introduced later in Section 3.4, and does not require roles to be specified by hand.

The reward structure is sparse and unforgiving: the reward function $R$ remains zero throughout an episode unless agents successfully visit all $L$ checkpoints in order, at which point they receive a positive reward. This all-or-nothing reward structure, combined with the sequential coordination requirement, creates a significant exploration challenge. We call $L$ the task's *coordination complexity*. The ordered checkpoint sequence should be understood as a stylised path through a more general coordination structure. In larger environments, checkpoints may form a graph rather than a single chain: a later checkpoint may be reachable through multiple predecessor checkpoints or through several alternative routes. Our theory isolates one feasible successful path through this graph. Multiple paths can improve constants by giving agents more opportunities to progress, but they do not remove the failure mode when progress along a selected path repeatedly requires revisiting earlier bottlenecks or coordination states.

Some checkpoints may be *hidden*—physically unreachable until earlier checkpoints unlock access. In the pressure-plate example, the second checkpoint $s^*_{(2)}$ might be locations inside a room that only becomes accessible after the door opens. To formalise this, we define the *frontier* $\mathcal{F}_k$ as the set of all states any agent

has visited up to episode $k$. A checkpoint is *unknown* if its component locations have not yet been discovered (that is, are not in $\mathcal{F}_k$). The frontier expands as agents explore newly accessible regions.

To analyse coordination success, we define a binary indicator for checkpoint achievement. For checkpoint $s^*_{(\ell)}$, the success indicator at time $t$ is

$$z_{t,\ell} = \mathbb{I}\left(s^1_t = s^*_{1,\ell} \wedge \ldots \wedge s^N_t = s^*_{N,\ell}\right).$$

This captures the all-or-nothing nature of coordination: partial success provides no reward and makes no progress. An episode *hits* checkpoint $\ell$ if $z_{t,\ell} = 1$ for at least one time step.

Our analysis uses two key metrics. Suppose that checkpoints $1, \ldots, \ell - 1$ have already been completed in order. The *next checkpoint* is then $s^*_{(\ell)}$, and $p_k$ denotes the probability that episode $k$ reaches this next uncompleted checkpoint. The expected hitting time $\tau_L$ measures the number of episodes required to complete the full ordered sequence of $L$ checkpoints:

$$p_k = \Pr(\text{episode } k \text{ hits the next uncompleted checkpoint}), \tag{1}$$

$$\tau_L = \mathbb{E}[\text{number of episodes until checkpoints } 1, \ldots, L \text{ are all completed}]. \tag{2}$$

Thus, $p_k$ captures local progress to the current target checkpoint, whereas $\tau_L$ captures overall task-completion efficiency. The hitting time $\tau_L$ is our primary measure of exploration strategy effectiveness, as it captures the expected number of episodes needed to solve the entire sequential coordination task.

### 3.3 Novelty-based exploration

To encourage exploration in sparse-reward settings, agents augment the extrinsic reward with an intrinsic bonus based on state novelty. Let $N^i_k(s_i)$ be agent $i$'s visit count to local state $s_i$ before episode $k$, and let $N^i_{e,k}(s_i)$ be the visit count within the current episode $k$. We consider three bonus schemes:

- **Lifelong bonus:** $r^i_{\text{life},k}(s_i) = [N^i_k(s_i) + 1]^{-1/2}$. This bonus permanently devalues states as they become familiar over the entire course of training.

- **Episodic bonus:** $r^i_{\text{epi},k}(s_i) = [N^i_{e,k}(s_i) + 1]^{-1/2}$. This bonus resets at the start of each episode, refreshing curiosity for all states.

- **Hybrid bonus:** $r^i_{\text{hyb},k}(s_i) = r^i_{\text{life},k}(s_i) \cdot r^i_{\text{epi},k}(s_i)$. This multiplicative combination gates the intrinsic reward by both temporal scales: a state receives a large bonus only when it is globally underexplored and also novel within the current episode. The lifelong term preserves a direction toward globally underexplored regions, while the episodic term discourages repeated within-episode cycling through the same local states.

These bonuses shape exploration behaviour differently. Lifelong bonuses push agents towards unexplored regions but may discourage revisiting critical coordination points. Episodic bonuses maintain consistent exploration patterns but lack memory across episodes. The hybrid approach aims to balance these trade-offs. We adopt a multiplicative rather than additive combination because the two forms induce different exploration biases. In an additive bonus,

$$r_{\text{add}} = \alpha r_{\text{life}} + \eta r_{\text{epi}},$$

the episodic term can dominate once the lifelong term decays, causing the method to behave similarly to episodic-only exploration and weakening the global discovery gradient. In contrast, multiplication acts as an "AND" gate: a state receives a high bonus only when it is both globally underexplored and episodically novel. This preserves the directionality of lifelong novelty while using episodic novelty to suppress repeated within-episode revisits (as also discussed by Henaff et al. (2023)). For completeness, we also report additive results in the experimental section.

The effectiveness of these strategies is measured by the coordination success probability $p_k$ and the expected hitting time $\tau_L$ as defined above. In the next subsection, we introduce a set of stylised assumptions that link novelty bonuses to visitation probabilities and allow us to analyse how task structure affects the guarantees we can provide for each scheme.

### 3.4 Modelling exploration–coordination trade-offs

Our theoretical results are derived within a stylised model of decentralised MARL in which intrinsic bonuses shape exploration but do not fully determine the learning rule. Instead of committing to a particular algorithm, we work under seven assumptions (A1–A7) that summarise the qualitative regularities we require: (i) novelty bonuses must influence visitation patterns in a monotone way, (ii) the environment may contain bottlenecks but is solvable within the horizon, and (iii) episodic bonuses limit within-episode revisits. Formal versions of A1–A7 are given in Appendix A; here we summarise their content and scope.

These assumptions should be read as sufficient conditions for the theoretical bounds, not as universal properties of all decentralised MARL tasks. They are most directly matched by the controlled GridWorld checkpoint tasks, where reachability, bottlenecks, and visit counts can be measured explicitly. In Overcooked and Star-Craft II, we use the theory as a qualitative lens for analogous bottleneck and revisit-pressure phenomena, rather than claiming that every formal assumption holds exactly.

**Behavioural assumptions (A1–A2).** The first pair of assumptions links intrinsic rewards to state visitation.

*Assumption A1 (fixed policy snapshot).* For the purposes of the theoretical analysis we consider a frozen snapshot of training in which the joint policy $\pi = (\pi_1, \ldots, \pi_N)$ is stationary across episodes. All expectations and visit-count bounds in this section are taken with respect to the Markov chain induced by this fixed policy. This is the standard "frozen policy" device used in the analysis of slowly updating RL algorithms: learning continues in practice, but the time-scale separation allows us to analyse exploration under a fixed policy snapshot.

*Assumption A2 (exploration–mixing).* For each agent $i$, let $\mathcal{S}^i$ denote its finite local state space and let $\mathcal{S}^i_{\text{reach}}(k) \subseteq \mathcal{S}^i$ denote the set of local states reachable from the current start distribution in episode $k$. Define the normalised intrinsic bonus

$$\bar{r}^i_k(s) = \frac{r^i_k(s)}{\sum_{u \in \mathcal{S}^i} r^i_k(u)}.$$

There exist multiplicative comparison constants $0 < \lambda_i \leq 1 \leq \kappa_i < \infty$ such that, for every episode $k$ prior to task completion and every $s \in \mathcal{S}^i_{\text{reach}}(k)$,

$$\lambda_i \, \bar{r}^i_k(s) \leq d^i_k(s) \leq \kappa_i \, \bar{r}^i_k(s),$$

where $r^i_k$ is the intrinsic bonus at the beginning of episode $k$, and $d^i_k(s)$ is the probability that agent $i$ visits $s$ at least once during that episode. The denominator above is the normalisation term; $\lambda_i$ and $\kappa_i$ are not normalisation constants, but multiplicative constants comparing bonus mass to visitation probability. No assumption is made about states that are not reachable from the current start distribution.

**Geometric and structural assumptions (A3–A6).** The next assumptions concern the environment and task structure.

*Assumption A3 (bounded cross-agent correlation).* Joint visitation probabilities for a given joint state are bounded above and below by the product of marginal visitation probabilities, up to explicit multiplicative constants. Specifically, there exist constants $0 < \rho_{\min} \leq 1 \leq \rho_{\max} < \infty$ such that, for every joint state $s = (s^1, \ldots, s^N)$ and episode $k$,

$$\rho_{\min} \prod_{i=1}^N d^i_k(s^i) \leq \Pr(\exists t \leq H : \ s_t = s \text{ in episode } k) \leq \rho_{\max} \prod_{i=1}^N d^i_k(s^i).$$

Here $d_k^i(s^i)$ denotes the probability that agent $i$ visits $s^i$ at least once during episode $k$. This rules out degenerate cases in which agents are forced to move in perfect lock-step or perfect anti-coordination, but still allows substantial interaction through blocking and emergent coordination.

A2 and A3 are behavioural regularity assumptions. They are not guaranteed by the bonus definition alone: they depend on the learning algorithm, policy class, and environment dynamics. We include these assumptions explicitly to make clear which behavioural link is required for the novelty bonus to imply a coordination guarantee. In tabular domains they can be checked empirically by comparing bonus-ranked states with observed visitation frequencies for A2, and by comparing joint checkpoint-hit probabilities with products of marginal visitation probabilities for A3. In Section E, we construct a simple grid-world setting to empirically check A2 and A3.

*Assumption A4 (geometric revisit exponent).* The geometry of the map may force agents to traverse early checkpoints or bottleneck states many times while searching for later checkpoints. We assume that the cumulative visit count of the first-checkpoint components grows at most polynomially with the number of episodes, with exponent $\beta_i \geq 0$ and constant $c_i$ for agent $i$:

$$N_k^i(s_{i,1}^*) \leq c_i k^{\beta_i}.$$

The exponent $\beta_i$ summarises revisit pressure for agent $i$, while $c_i$ is an agent-specific scaling constant controlling the baseline magnitude of revisit growth at the first checkpoint component. We use $\beta = \max_i \beta_i$ as a conservative task-level value.

This quantity is directly relevant to lifelong novelty: if $N_k \approx k^\beta$, then a count-based lifelong bonus decays as $k^{-\beta/2}$, and the corresponding joint lower bound for $N$ agents decays as $k^{-\beta N/2}$. Thus $\beta$ does not describe all forms of task difficulty; it isolates the geometric component that specifically harms lifelong novelty by repeatedly depleting task-critical states.

*Assumption A5 (frontier exploration).* States that have never been visited carry maximal intrinsic bonus. This is satisfied by count-based exploration schemes by construction and is a natural regularity condition for novelty estimators more generally: genuinely new regions remain attractive.

*Assumption A6 (gated reachability).* The ordered checkpoint sequence is feasible within the episode horizon in a staged sense. Let $h_1, \ldots, h_L$ be positive integer stage budgets with $\sum_{\ell=1}^{L} h_\ell \leq H$, and define $T_0 = 0$ and $T_\ell = \sum_{m=1}^{\ell} h_m$. Let $E_{\ell,k}$ denote the event that checkpoint $\ell$ is reached during its allocated stage in episode $k$:

$$E_{\ell,k} = \left\{ \exists t \in \{T_{\ell-1} + 1, \ldots, T_\ell\} : \ s_t = s_{(\ell)}^* \right\}.$$

There exist constants $q_1, \ldots, q_L \in (0, 1]$ such that, for each $\ell = 1, \ldots, L$, conditional on checkpoints $1, \ldots, \ell-1$ having been reached within their allocated prefix budget, checkpoint $\ell$ remains stage-reachable and satisfies

$$\Pr(E_{\ell,k} \mid E_{1,k} \cap \cdots \cap E_{\ell-1,k}) \geq q_\ell \, \rho_{\min} \prod_{i=1}^{N} d_k^i(s_{i,\ell}^*),$$

with the conditioning event for $\ell = 1$ interpreted as the whole probability space. We write

$$q_{\text{gate}} := \min_{\ell=1,\ldots,L} q_\ell > 0.$$

This assumption is a feasibility condition on an executable within-horizon staged path, rather than a generic lower bound on the horizon length.

A6 holds most directly in our controlled GridWorld checkpoint tasks, where doors and rooms create explicit staged reachability. In Overcooked and StarCraft II, the assumption should be interpreted qualitatively: these domains contain repeated handoff points, bottlenecks, or tactical configurations that create analogous revisit pressure, but we do not claim that the formal gated-reachability assumption holds exactly in every state.

**Episodic-memory assumption (A7).** The final assumption captures the qualitative effect of episodic bonuses.

*Assumption A7 (per-episode revisit cap).* When an episodic component is present, there exists a constant $C_{\text{epi}}$ that does not scale with the horizon $H$ such that, within a single episode, an agent visits any given state at most $C_{\text{epi}}$ times. This formalises the observation that episodic bonuses rapidly downweight states after one or two visits, thereby discouraging repeated back-and-forth behaviour.

Taken together, A1–A7 describe a broad class of decentralised MARL settings in which intrinsic rewards influence policies in a monotone way, the environment may contain geometric bottlenecks but remains solvable, and episodic bonuses (when present) suppress excessive within-episode revisits. The subsequent lemmas should therefore be interpreted as sufficient-condition guarantees within this regime, rather than as claims tied to a single algorithm or benchmark.

### 3.5 Main results

Having established the formal framework, we now analyse when different exploration strategies enjoy provable guarantees. The key quantities are the coordination complexity $L$ and the geometric revisit pressure $\beta$: larger values force agents to repeatedly traverse earlier checkpoints whilst searching for later ones, creating a tension between exploration progress and maintaining coordination.

When agents must find multiple coordination points sequentially ($L > 1$), they typically have to pass through earlier checkpoints many times while attempting to discover later ones. Under lifelong novelty, this repeated traversal gradually depletes the bonus at these critical locations. Over time, the incentive to return can become very weak, creating a form of coordination "death spiral" in which exploration towards new regions undermines the ability to re-synchronise at earlier checkpoints.

Our analysis centres on the geometric revisit exponent $\beta$ introduced in Assumption A4, which quantifies how severely the environment forces checkpoint revisitation. Within this stylised setting, we address three questions:

1. How can we bound the coordination probability under different bonus schemes?

2. How do these bounds depend on task complexity ($L$) and revisit pressure ($\beta$)?

3. Under what conditions do standard novelty mechanisms cease to offer finite-time guarantees?

We now present our main theoretical results, starting with the simplest scheme and building towards the hybrid case. Proofs are given in Appendix B.

**Lemma 1 (Episodic bonus provides a constant floor)** *Under Assumptions 2 and 3, the episodic bonus guarantees a time-uniform lower bound on the single-episode success probability for any reachable checkpoint:*

$$p_k^{\text{epi}} \geq \rho_{\min} \prod_{i=1}^{N} \frac{\lambda_i}{|\mathcal{S}^i|} =: p_{\min}^{\text{epi}} > 0 \quad \text{for all episodes } k.$$

*In particular, the first hitting time $\tau_{\text{epi}}$ has finite expectation bounded by a constant depending only on $\rho_{\min}$, $\lambda_i$ and $|\mathcal{S}^i|$.*

**Proof sketch.** At the start of each episode, episodic visit counts are reset, so the episodic bonus is $r_{\text{epi},k}^i(s) = 1$ for all states. The total bonus mass for agent $i$ is therefore $|\mathcal{S}^i|$, and Assumption 2 implies $d_k^i(s) \geq \lambda_i / |\mathcal{S}^i|$ for any state $s$.

Applying Assumption 3 to the joint checkpoint $s_{(1)}^*$ then yields $p_k^{\text{epi}} \geq \rho_{\min} \prod_i \lambda_i / |\mathcal{S}^i|$, uniformly in $k$. A geometric argument based on this time-uniform lower bound implies that the expected hitting time is finite and bounded in terms of $\rho_{\min}$, $\lambda_i$ and $|\mathcal{S}^i|$, where $|\mathcal{S}^i|$ denotes the size of agent $i$'s local state space. See Appendix B for the full argument. $\square$

This result captures the fundamental strength of episodic bonuses: because novelty estimates are reset at the start of each episode, all states begin equally attractive, and the probability of synchronising at a given checkpoint does not decay with time. The bound depends on how strongly bonuses influence behaviour ($\lambda_i$), how much agents interfere ($\rho_{\min}$), and the size of each local state space, but not on the episode index $k$. The drawback is that this time-uniformity is achieved by discarding cross-episode memory: episodic bonuses alone cannot systematically bias exploration towards as-yet-unseen regions where later checkpoints may be located.

**Lemma 2 (Lifelong bonus performance depends on $\beta$)** *Under Assumptions 2, 3, 4 and 1, define*

$$C_{\min} \;=\; \rho_{\min} \prod_{i=1}^{N} \frac{\lambda_i}{|\mathcal{S}^i|} \, (1+c_i)^{-1/2}.$$

*Then, for every episode $k$ before the first success,*

$$p_k^{\mathrm{life}} \;\geq\; C_{\min} \, k^{-\beta N/2}.$$

*If $\frac{\beta N}{2} < 1$ (equivalently, $\beta N < 2$), then the lifelong exploration scheme reaches the first coordination checkpoint in finite expected episodes:*

$$\mathbb{E}[\tau_{\mathrm{life}}] \;<\; \infty.$$

**Proof sketch.** For the first checkpoint component $s_{i,1}^*$ of agent $i$, Assumption 4 bounds the cumulative visit count as $N_k^i(s_{i,1}^*) \leq c_i k^{\beta_i}$.

The lifelong bonus is $r_{\mathrm{life},k}^i(s) = (N_k^i(s) + 1)^{-1/2}$, so at the checkpoint we obtain $r_{\mathrm{life},k}^i(s_{i,1}^*) \geq (1 + c_i)^{-1/2}k^{-\beta_i/2}$. Assumption 2 converts this into a lower bound on the visitation probability $d_k^i(s_{i,1}^*) \geq \frac{\lambda_i}{|\mathcal{S}^i|}(1 + c_i)^{-1/2}k^{-\beta_i/2}$. Combining across agents via Assumption 3 yields $p_k^{\mathrm{life}} \geq C_{\min}k^{-\sum_i \beta_i/2} \geq C_{\min}k^{-\beta N/2}$, where $\beta = \max_i \beta_i$ and $C_{\min}$ is as in the lemma.

To control the hitting time, we consider the tail probability $\Pr(\tau_{\mathrm{life}} > k)$ and bound it using $\Pr(\tau_{\mathrm{life}} > k) \leq \exp(-\sum_{t=1}^{k} p_t^{\mathrm{life}})$. When $\beta N/2 < 1$, the polynomial lower bound on $p_t$ implies that $\sum_{t=1}^{k} p_t^{\mathrm{life}}$ grows at least as $k^{1-\beta N/2}$, which gives a sub-exponential tail for $\Pr(\tau_{\mathrm{life}} > k)$ and hence a finite first moment. For larger $\beta N$, this argument no longer produces a finite bound. The full argument is given in Appendix B. □

The lower bound in Lemma 2 shows that the *guaranteed* success probability we can derive decays polynomially with the episode index, with exponent proportional to $\beta N/2$. As either the revisit pressure $\beta$ or the number of agents $N$ increases, this worst-case guarantee becomes weaker. For $\beta N < 2$ it is still strong enough to imply a finite expected hitting time for the *first* checkpoint. When $\beta N$ is larger, our sufficient conditions no longer guarantee finite expected time, and the bound becomes uninformative. In practice we observe that lifelong-only exploration degrades sharply in high-$\beta$ settings (Section 5), which is consistent with this dependence on revisit pressure.

**Lemma 3 (Hybrid bonus ensures stability and discovery)** *Under Assumptions 2, 3, 5 and 1, the hybrid bonus $r_{\mathrm{hyb}} = r_{\mathrm{life}} \cdot r_{\mathrm{epi}}$ inherits key properties from both components:*

1. *Inward stability. For any known checkpoint and any finite training horizon $K_{\max}$, there exists a constant $\eta > 0$ (depending on $H$, $K_{\max}$ and $\lambda_i$) such that the coordination probability satisfies*

$$p_k^{\mathrm{hyb}} \;\geq\; \rho_{\min} \, \eta^N \quad \text{for all} \quad k \leq K_{\max}.$$

2. *Outward discovery. For any reachable state $s \notin \mathcal{F}_k$ (outside the current frontier), the probability of visiting $s$ in episode $k$ is lower-bounded by a positive constant that depends only on $\lambda_i$ and $|\mathcal{S}^i|$.*

**Proof sketch.** For inward stability, consider a known checkpoint. Within a frozen-policy window of length $K_{\max}$, each agent can visit any given state at most $H$ times per episode, so the total number of visits is at

most $K_{\max}H$. The lifelong and episodic bonuses are of the form $(N+1)^{-1/2}$, hence at that checkpoint we have $r^i_{\mathrm{epi},k} \geq 1/\sqrt{H+1}$ and $r^i_{\mathrm{life},k} \geq 1/\sqrt{K_{\max}H+1}$ for all $k \leq K_{\max}$. Their product is therefore bounded below by a constant independent of $k$, and Assumptions 2 and 3 yield a constant lower bound $\rho_{\min}\eta^N$ on the single-episode coordination probability.

For outward discovery, if a state $s$ has never been visited by agent $i$ then Assumption 5 gives $N^i_k(s) = N^i_{e,k}(s) = 0$, so $r^i_{\mathrm{hyb},k}(s) = 1$. All bonuses are at most 1, hence the total bonus mass is at most $|\mathcal{S}^i|$, and Assumption 2 implies $d^i_k(s) \geq \lambda_i/|\mathcal{S}^i|$. Applying Assumption 3 again yields a positive lower bound on the joint probability of visiting $s$. See Appendix B for the full details. $\qquad\square$

The multiplicative combination $r_{\mathrm{hyb}} = r_{\mathrm{life}} \cdot r_{\mathrm{epi}}$ achieves both properties within a finite frozen-policy window. The episodic component does not restore a large reward after the lifelong term has been fully depleted. Instead, it changes the trajectory distribution by discouraging repeated within-episode cycling, which limits how quickly task-critical states are depleted. The lifelong component still preserves maximal bonus for truly unvisited states. Thus the hybrid method combines finite-window re-coordination stability with outward discovery, under the stated assumptions.

**Lemma 4 (Hybrid finite-window sequential bound with episodic cap)** *Fix a frozen-policy analysis window consisting of episodes $k_0 + 1, \ldots, k_0 + K_{\max}$, each of horizon $H$. Let $q_{\mathrm{gate}}$ be the staged-reachability constant from Assumption 6. Suppose that the lifelong counts of all checkpoint components at the start of the window are bounded by*

$$B_{\max} \geq \max_{i,\ell} N^i_{k_0}(s^*_{i,\ell}).$$

*Under Assumptions 1–6 and Assumption 7, define*

$$\bar{q}^\dagger := \rho_{\min}\left(B_{\max} + K_{\max}C_{\mathrm{epi}} + 1\right)^{-N/2} \prod_{i=1}^N \frac{\lambda_i}{|\mathcal{S}^i|}.$$

*Then, conditionally on any history before an episode in the window, that episode completes the ordered checkpoint sequence with probability at least*

$$(q_{\mathrm{gate}}\bar{q}^\dagger)^L.$$

*Consequently, conditional on no earlier task completion,*

$$\Pr\left(\tau^{\mathrm{hyb}}_L \leq k_0 + K_{\max} \mid \tau^{\mathrm{hyb}}_L > k_0\right) \geq 1 - \left(1 - (q_{\mathrm{gate}}\bar{q}^\dagger)^L\right)^{K_{\max}}.$$

*If the same per-episode lower bound is maintained after $k_0$ until the first successful episode, then the residual hitting time satisfies*

$$\mathbb{E}\left[\tau^{\mathrm{hyb}}_L - k_0 \mid \tau^{\mathrm{hyb}}_L > k_0\right] \leq (q_{\mathrm{gate}}\bar{q}^\dagger)^{-L}.$$

*In particular, when $k_0 = 0$ and the bound is maintained from the start of training, this gives*

$$\mathbb{E}\left[\tau^{\mathrm{hyb}}_L\right] \leq (q_{\mathrm{gate}}\bar{q}^\dagger)^{-L}.$$

**Proof sketch.**

By Assumption 7, within the $K_{\max}$-episode window, any checkpoint component $s^*_{i,\ell}$ is visited by agent $i$ at most $K_{\max}C_{\mathrm{epi}}$ additional times. Since the count at the start of the window is at most $B_{\max}$, the lifelong part of the hybrid bonus satisfies

$$r^i_{\mathrm{life},k}(s^*_{i,\ell}) \geq (B_{\max} + K_{\max}C_{\mathrm{epi}} + 1)^{-1/2}$$

for every checkpoint $\ell$ and every episode in the window. At the beginning of each episode, episodic counts are reset, so the episodic multiplier is 1. Thus the same lower bound applies to the hybrid bonus at the start of the episode.

Since all count-based bonuses are at most 1, the total bonus mass is at most $|\mathcal{S}^i|$, and Assumption 2 gives

$$d_k^i(s_{i,\ell}^*) \geq \frac{\lambda_i}{|\mathcal{S}^i|}(B_{\max} + K_{\max}C_{\mathrm{epi}} + 1)^{-1/2}.$$

Combining the agents through Assumption 3 gives

$$\rho_{\min}\prod_{i=1}^{N} d_k^i(s_{i,\ell}^*) \geq \bar{q}^\dagger.$$

Assumption 6 then ensures that, conditional on the earlier checkpoints having been reached within their allocated stage budgets, the probability of reaching checkpoint $\ell$ in its allocated stage is at least $q_{\mathrm{gate}}\bar{q}^\dagger$. Therefore the probability of completing all $L$ checkpoints in one episode is at least

$$(q_{\mathrm{gate}}\bar{q}^\dagger)^L.$$

Applying this conditional lower bound over the $K_{\max}$ episodes gives

$$\Pr\left(\tau_L^{\mathrm{hyb}} \leq k_0 + K_{\max} \mid \tau_L^{\mathrm{hyb}} > k_0\right) \geq 1 - \left(1 - (q_{\mathrm{gate}}\bar{q}^\dagger)^L\right)^{K_{\max}}.$$

If the same per-episode lower bound is maintained until success, the usual geometric waiting-time argument gives the residual expectation bound. The detailed argument is given in Appendix B. □

Lemma 4 explains why the hybrid scheme is particularly well-suited to sequential coordination tasks. The crucial ingredient is the effective per-episode revisit cap in Assumption 7: whereas a lifelong-only agent might visit a state up to $H$ times in a single episode, the episodic component ensures that the number of revisits is bounded by $C_{\mathrm{epi}} \ll H$. Over a $K_{\max}$-episode window, this changes the relevant count-dependent term from roughly $B_{\max} + K_{\max}H$ to $B_{\max} + K_{\max}C_{\mathrm{epi}}$, slowing the depletion of task-critical states under the hybrid bonus. The additional factor $q_{\mathrm{gate}}$ records the staged feasibility of the checkpoint path. It is independent of the novelty mechanism and captures the probability cost of executing the ordered within-horizon transitions once the relevant checkpoint components remain reachable.

This should not be interpreted as an asymptotic non-decay guarantee. If the lifelong component at a state has already become arbitrarily small, multiplication by an episodic term cannot make the absolute hybrid bonus large again. The benefit of the episodic component is preventive: by reducing repeated within-episode revisits, it slows the depletion of coordination-critical states over the finite training window in which the policy is being learned.

Comparing Lemmas 2 and 4, the hybrid bound replaces the worst-case within-window revisit scale $H$ by the smaller effective scale $C_{\mathrm{epi}}$. When $B_{\max}$ is not dominant, this corresponds to an improvement in the checkpoint-level lower-bound term on the order of $(H/C_{\mathrm{epi}})^{N/2}$. Unlike the lifelong-only lower bound in Lemma 2, the finite-window lower bound in Lemma 4 holds for any coordination depth $L$ and does not depend on the geometric parameter $\beta$. If the same per-episode lower bound is maintained until success, the finite-window argument immediately yields the stated residual expected hitting-time bound.

**Exploration regimes.** The three lemmas above highlight two qualitatively different regimes for lifelong exploration:

- **Low revisit pressure ($\beta N < 2$).** The lower bound in Lemma 2 is strong enough to guarantee finite expected time to reach the first checkpoint. Episodic bonuses retain a time-uniform lower bound on reachable checkpoints, and the hybrid scheme retains a finite-window lower bound while preserving outward discovery.

- **High revisit pressure ($\beta N \geq 2$).** Our sufficient conditions no longer guarantee finite expected time for lifelong exploration, and the bound becomes increasingly weak as $\beta N$ grows. In contrast, the episodic and hybrid schemes retain lower bounds on re-coordination probability that are independent of $\beta$, with the hybrid guarantee stated over finite frozen-policy windows.

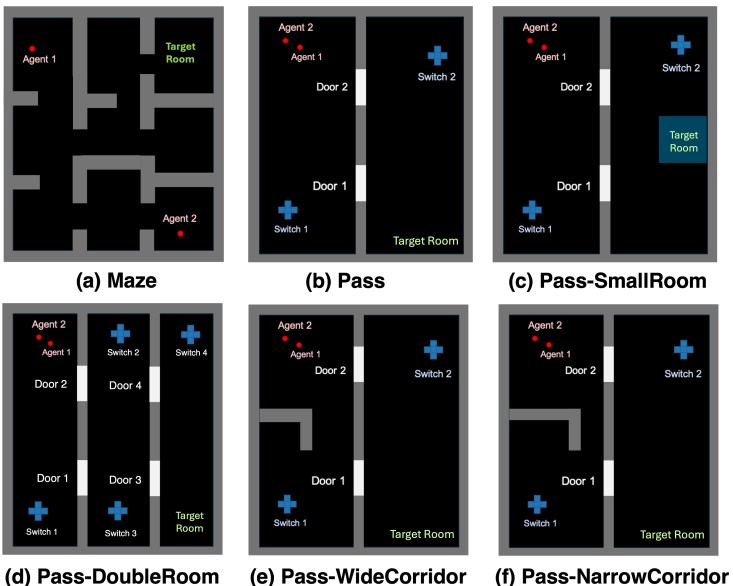

Figure 1: Grid environments with varying coordination complexity and geometric pressure. (a) Maze: $L = 1$, agents must reach the target room together. (b) Pass: $L = 2$, agents coordinate at two switch–door pairs. (c) Pass-SmallRoom: $L = 3$, where the third coordination checkpoint is the event in which both agents arrive together in the target room. (d) Pass-DoubleRoom: $L = 4$, requires four sequential coordinations. (e) Pass-WideCorridor: $L = 2$, a wide corridor reduces revisit pressure. (f) Pass-NarrowCorridor: $L = 2$, a narrow corridor increases geometric pressure ($\beta$).

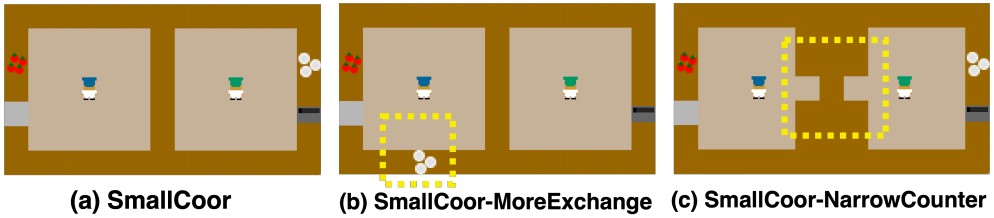

Figure 2: Overcooked environments with varying coordination demands. Agents must cooperate to prepare and serve soup for reward. (a) SmallCoor: $L = 2$, broad counter access for exchanging items. (b) SmallCoor-MoreExchange: $L = 3$, dishes relocated to the left room, adding a third exchange requirement (yellow box). (c) SmallCoor-NarrowCounter: high $\beta$, a narrow exchange area (yellow box) forces repeated visits to the same coordination point.

The hybrid scheme combines these advantages: it enjoys a $\beta$-independent finite-window lower bound for any coordination depth $L$ (Lemma 4) whilst still preserving the outward exploratory drive of lifelong bonuses. Our experiments in Section 5 vary both the geometric parameter $\beta$ and the coordination complexity $L$ across multiple environments, and exhibit degradation patterns for lifelong-only exploration, and stability for hybrid exploration, that are consistent with these theoretical guarantees.

**Extension to communication.** Although our theoretical analysis assumes agents operate without explicit communication, the framework readily extends to settings where agents share limited information. Communication primarily affects three aspects of the bounds: it can increase the exploration constants $\lambda_i$ (Assumption 2) by improving the effectiveness of novelty signals, increase $\rho_{\min}$ (Assumption 3) by reducing conflicts, and shorten the fixed-policy window $K_{\max}$ (Assumption 1) through faster convergence. Crucially, the geometric revisit exponent $\beta$ (Assumption 4) is determined by map structure and is unaffected by communication.

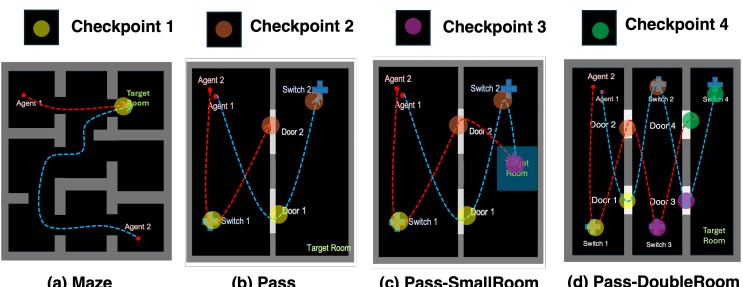

Figure 3: Coordination complexity in GridWorld. Different coloured circles indicate distinct checkpoints. To complete the task, agents must sequentially occupy all checkpoints simultaneously. In `Maze`, there is a single checkpoint ($L = 1$): both agents must reach the target room at the same time. In `Pass-DoubleRoom`, there are four checkpoints ($L = 4$) that must be completed in sequence.

These changes tighten the constant $\bar{q}^{\dagger}$ in Lemma 4 whilst preserving the qualitative picture: the lifelong guarantee continues to depend on $\beta N$ through Lemma 2, whereas the episodic and hybrid guarantees remain $\beta$-independent and retain finite-window lower bounds on re-coordination probability. In this sense, the analysis provides a common foundation for both zero-communication and communication-enhanced decentralised settings.

## 4 Experimental setup

We evaluate our theoretical predictions across three environments: GridWorld, Overcooked, and StarCraft II. Unless otherwise specified, we use IPPO (De Witt et al., 2020) as the base learning algorithm and train agents with different intrinsic bonus schemes. All environments are configured as sparse-reward tasks: agents receive extrinsic reward only upon successful task completion, making coordination discovery particularly challenging.

**GridWorld.** This two-agent environment requires coordination via switches and doors to reach target rooms. Each switch–door mechanism or simultaneous room entry defines a coordination checkpoint, allowing precise control over coordination complexity ($L$) and geometric revisit pressure ($\beta$). We evaluate six maps (Fig. 1): `Maze` ($L = 1$), `Pass` ($L = 2$), `Pass-SmallRoom` ($L = 3$), and `Pass-DoubleRoom` ($L = 4$) test increasing coordination complexity (Figure 3). To isolate geometric effects, we compare `Pass-WideCorridor` and `Pass-NarrowCorridor` (both $L = 2$): the narrow corridor forces frequent traversal of the same bottleneck cells, resulting in higher revisit pressure.

To empirically quantify geometric pressure, we select, for each map, a small region in the hallway or doorway that agents must repeatedly cross to complete the task. For training epoch $k$, we record the mean visit count $N_k$ of each cell in this region and fit a log–log relationship

$$\log N_k = \beta \log k + \alpha.$$

A larger slope $\beta$ indicates that the visit count for that bottleneck grows more rapidly with training, corresponding to stronger revisit pressure and faster decay of lifelong novelty. The fitted slopes for `Pass`, `Pass-WideCorridor`, and `Pass-NarrowCorridor` are 1.02, 1.05, and 1.89, respectively (Figure 4), indicating substantially higher geometric pressure in the narrow-corridor variant.

**Overcooked.** This environment requires two agents to cooperate continuously to prepare and serve food. We evaluate three layouts with varying coordination demands (Fig. 2). `SmallCoor` provides agents with broad access to a shared counter, requiring an exchange of tomatoes and cooked soup ($L = 2$). `SmallCoor-MoreExchange` (ME) relocates dishes to the left room, adding a third exchanged item ($L = 3$). `SmallCoor-NarrowCounter` (NC) restricts exchanges to a narrow counter segment, increasing revisit pressure $\beta$ as agents must repeatedly return to the same coordination spot to pass items.

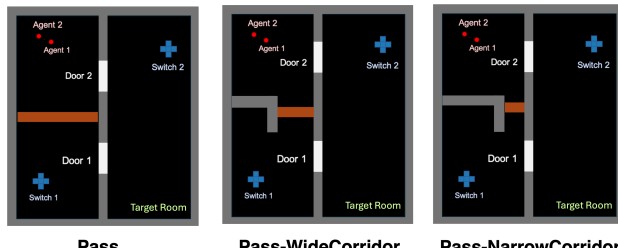

**Pass**      **Pass-WideCorridor**      **Pass-NarrowCorridor**

Figure 4: Geometric pressure in GridWorld. For `Pass`, `Pass-WideCorridor` and `Pass-NarrowCorridor`, we select a short hallway or doorway region that an agent must backtrack through to finish the task. For each training epoch $k$, we record the mean visit count $N_k$ over cells in this region and fit $\log N_k = \beta \log k + \alpha$. A larger slope $\beta$ means that visits to the bottleneck grow more rapidly with episodes, reflecting stronger revisit pressure and quicker loss of lifelong novelty. The fitted slopes are 1.02, 1.05 and 1.89 for `Pass`, `Pass-WideCorridor` and `Pass-NarrowCorridor`, respectively.

**StarCraft II.** We test three standard combat scenarios in this real-time environment. Success requires continuous sequences of coordinated actions (positioning, target selection, retreating), yielding high coordination complexity ($L > 1$) at the level of temporal action sequences rather than explicit spatial checkpoints. In `2m_vs_1z`, two Marines must defeat a Zealot using a "kiting" strategy—repeatedly moving and firing—which forces agents through similar tactical configurations and induces high revisit pressure. `3m` and `8m` feature symmetric battles requiring coordinated focus-firing and movement, similarly inducing high $\beta$ as agents repeatedly converge on key battlefield areas. Further environment details are provided in Appendix C.

## 5 Experimental results

Our experiments test the core predictions of our theoretical framework under varying communication conditions. The theoretical analysis considers exploration dynamics without communication, in order to isolate fundamental effects, and we then evaluate whether the same qualitative patterns hold when agents can share information. We test three novelty schemes—*Lifelong only*, *Episodic only*, and *Combined* (our hybrid approach)—under three communication settings:

- **Local novelty only:** $r_{\text{int}}^i = u_t^i$ (each agent uses only its own novelty signal).

- **Summation of local novelty:** $r_{\text{int}}^i = u_t^i + \sum_{j \neq i} u_t^j$ (agents use the sum of all local novelty signals).

- **MACE:** $r_{\text{int}}^i = u_t^i + \sum_{j \neq i} u_t^j + \sum_{j \neq i} v_{i,j}$ (agents additionally receive an influence term $v_{i,j}$ as in Jiang et al. (2024)).

Here $u_t^i$ denotes agent $i$'s intrinsic novelty at time $t$ and $v_{i,j}$ is an influence term; further details are given in Appendix D. From the perspective of our bounds, communication primarily increases the exploration constants $\lambda_i$ and the correlation parameter $\rho_{\min}$, whilst leaving the geometric parameter $\beta$ unchanged, and thus tightens the constants in our analytical guarantees without changing their qualitative form. Empirically, we observe improvements consistent with this picture across all environments.

For implementation details, in GridWorld we use tabular, count-based novelty. Lifelong and episodic bonuses are given by

$$u_t^{i,\text{life}} = \frac{1}{\sqrt{N^i(s_t^i) + 1}}, \qquad u_t^{i,\text{epi}} = \mathbb{I}\big(N_e^i(s_t^i) = 1\big),$$

where $N^i(\cdot)$ counts total visits across all episodes and $N_e^i(\cdot)$ counts visits within the current episode. The hybrid intrinsic reward uses the multiplicative combination

$$u_t^{i,\text{comb}} = u_t^{i,\text{life}} \cdot u_t^{i,\text{epi}} = \frac{1}{\sqrt{N^i(s_t^i) + 1}} \mathbb{I}\big(N_e^i(s_t^i) = 1\big).$$

For continuous-state environments (Overcooked and StarCraft II), we use Random Network Distillation (RND) (Burda et al., 2019) to estimate lifelong novelty and a distance-to-episodic-memory term to estimate episodic novelty. Concretely, for agent $i$ at time $t$ with observation $s_t^i$, we define

$$u_t^{i,\text{life}} = \text{RND}(s_t^i), \qquad u_t^{i,\text{epi}} = \min_{j \in \{0,1,\ldots,t-1\}} d\big(\phi(s_t^i), \phi(s_j^i)\big),$$

and

$$u_t^{i,\text{comb}} = u_t^{i,\text{life}} \cdot u_t^{i,\text{epi}}.$$

Here $\phi(\cdot)$ is a feature encoder learned following Henaff et al. (2022): given transitions $(s_t^i, a_t^i, s_{t+1}^i)$, the encoder $\phi$ and a prediction head $g$ are trained to optimise

$$L(s_t^i, a_t^i, s_{t+1}^i; \phi, g) = -\log p\big(a_t^i \mid g\big(\phi(s_t^i), \phi(s_{t+1}^i)\big)\big),$$

so that $\phi$ focuses on information that is predictive of the agent's own actions and is less sensitive to uncontrollable noise. We also experimented with the full E3B intrinsic motivation (Henaff et al., 2022), but found that this simpler distance-based episodic term, combined with RND, yielded more stable performance in our multi-agent settings.

## 5.1 Analysis of GridWorld environments

GridWorld provides the most controlled test of our theory through systematic manipulation of coordination complexity ($L$) and geometric revisit pressure ($\beta$), and illustrates how lifelong exploration degrades as these factors increase.

In `Maze` ($L = 1$), all three strategies perform comparably well with or without communication (Figs. 5, 6), as expected when no checkpoint revisitation is required. As coordination complexity increases to `Pass` ($L = 2$), lifelong-only performance degrades, particularly without communication. This degradation accelerates in `Pass-SmallRoom` ($L = 3$) and `Pass-DoubleRoom` ($L = 4$), where lifelong-only fails across all seeds (Fig. 6).

Manipulating geometric pressure further confirms the theoretical dependence on $\beta$: `Pass-NarrowCorridor` forces more frequent checkpoint revisitation than `Pass-WideCorridor` and `Pass`, and lifelong-only performs correspondingly worse (Fig. 6). Across these scenarios, the combined approach maintains strong performance, demonstrating that incorporating an episodic component becomes increasingly important as either $L$ or $\beta$ increases. Figure 7 makes this comparison explicit by plotting final return as coordination complexity increases from `Maze` to `Pass-DoubleRoom`, and as revisit pressure increases from `Pass-WideCorridor` to `Pass-NarrowCorridor`. The degradation of lifelong-only exploration is strongest when either $L$ or $\beta$ is large, while the combined bonus remains substantially more stable.

Heat maps (Fig. 8) reveal the underlying mechanism: lifelong-only exploration produces patchy, desynchronised coverage as agents lose motivation to revisit depleted areas, whilst the combined approach maintains diverse, coordinated exploration patterns. With an episodic-only bonus, agents are encouraged to visit different local states within each episode, producing scattered trajectories and reducing repeated cycling. However, because episodic novelty resets at the start of every episode, it does not provide persistent cross-episode pressure toward globally underexplored regions. As a result, episodic-only exploration can perform well in single-checkpoint tasks such as `Maze`, and in compact layouts where later checkpoints are close to the start, but it struggles to systematically discover and stabilise later checkpoints in harder GridWorld chains such as `Pass`, `Pass-SmallRoom`, and `Pass-DoubleRoom`. In Figure 9, We added a checkpoint-discovery-rate diagnostic, measuring when agents first reach later checkpoints during training, but noted that a single hit does not necessarily mean the behavior has been immediately learned. Results show that Checkpoint 1 could be reached by all strategies in all environments within early training. For deeper checkpoints, Combined consistently achieves the highest hit rate and the fastest convergence across all environments, while Lifelong shows competitive hit rate in simpler environments (Pass, Pass-NarrowCorridor) but fail to reach the final checkpoints in the more complex Pass-DoubleRoom setting. For Episodic, in Pass and Pass-NarrowCorridor, it typically reaches only the first checkpoint and rarely progresses to the second. In Pass-DoubleRoom, a small number of seeds manage to reach intermediate checkpoints (second and third), but none succeed in

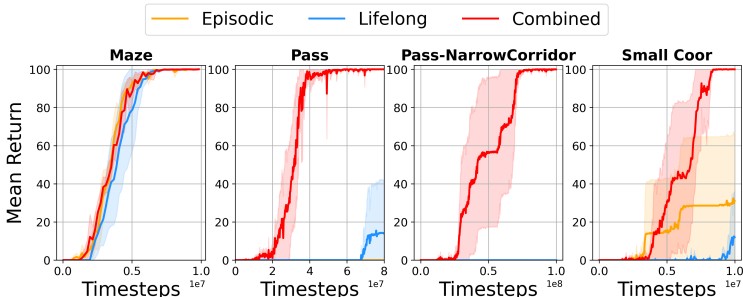

Figure 5: Learning curves comparing episodic, lifelong, and combined bonuses in the local novelty-only setting (no communication). Results show mean return over 7 seeds with 95% confidence intervals. Combined approach maintains strong performance across all environments while lifelong-only fails in high-complexity (Pass) and high-pressure (Pass-NarrowCorridor) scenarios. These conventions (7 seeds, 95% CI) apply to all subsequent figures.

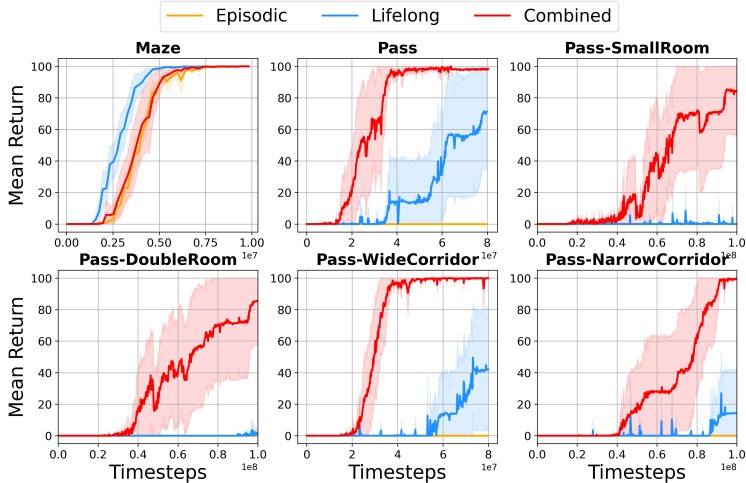

Figure 6: Learning curves of Episodic, Lifelong, and Combined Bonuses in Summation of Local Novelty setting across GridWorld variants.

reaching the final checkpoint. This confirms that the combined bonus discovers task-relevant coordination states more reliably than episodic-only or lifelong-only exploration.

For Table 1, we additionally test whether the best-performing method significantly outperforms the second-best method using a two-sided Mann–Whitney U test on final evaluation returns across the 7 random seeds. Given the small number of seeds, we view these tests as supportive rather than definitive, but they provide a useful non-parametric check under the highly discrete return distributions in these sparse-reward coordination tasks, where many runs end in either success or failure.

## 5.2 Generality across other domains

Our theoretical principles extend to more complex domains beyond tabular GridWorld. In StarCraft II's `2m_vs_1z`, the optimal "kiting" strategy—requiring Marines to repeatedly advance and retreat—creates high revisit pressure through emergent gameplay rather than explicit map design: agents must cycle through similar tactical configurations many times. As predicted, the lifelong-only scheme struggles to discover and stabilise this behaviour, while the combined approach reliably learns effective kiting policies (Fig. 10).

The Overcooked environments provide complementary evidence. In `SmallCoor-NC`, physical constraints create a narrow exchange region that acts as a coordination bottleneck, increasing revisit pressure $\beta$ at the

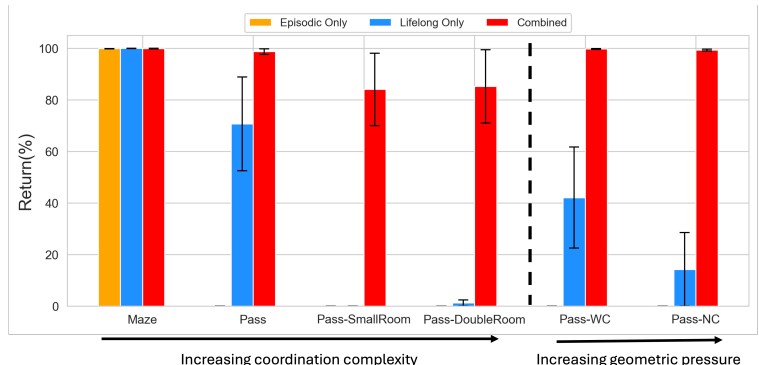

Figure 7: Final return of episodic-only, lifelong-only, and combined bonuses under controlled changes in coordination complexity and revisit pressure in GridWorld. The left group increases coordination complexity $L$ from `Maze` to `Pass-DoubleRoom`; the right group compares layouts with similar $L$ but different geometric revisit pressure, from `Pass-WideCorridor` to `Pass-NarrowCorridor`. Lifelong-only performance degrades as either $L$ or $\beta$ increases, while the combined bonus is more stable.

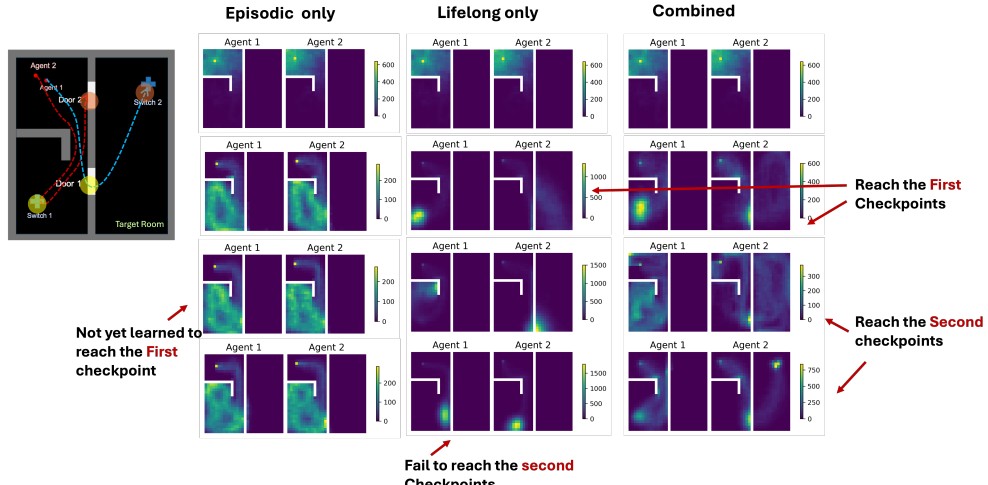

Figure 8: Visit counts in `Pass-NarrowCorridor` during training for episodic-only, lifelong-only, and combined agents. The top row shows visit counts at an early training epoch; the bottom row shows visit counts later in training. Brighter cells indicate higher visit counts. The episodic-only agent spreads visits across many states but lacks a persistent cross-episode drive toward the next room. The lifelong-only agent can reach the first coordination checkpoint, but its visits become patchy and de-synchronised as novelty at critical states is depleted. The combined agent maintains more diverse behaviour while preserving enough directionality to reach the second checkpoint and obtain the final reward.

handover point. In `SmallCoor-ME`, relocating dishes to a separate room adds a third exchange, increasing coordination complexity $L$. With communication (Summation and MACE settings), the combined approach outperforms lifelong-only exploration in both layouts (Figs. 10, 11); without communication, `SmallCoor-NC` is sufficiently challenging that the combined scheme does not reliably solve the task. This is not a contradiction of the theory: the bounds are sufficient-condition guarantees and depend on non-negligible exploration and synchronisation constants such as $\lambda_i$ and $\rho_{\min}$. In `SmallCoor-NC` with local novelty only, agents rarely discover that the narrow counter is the relevant handoff bottleneck, so the effective constants are extremely small. When novelty information is shared through Summation or MACE, the same layout becomes much easier, consistent with the view that communication improves these constants while leaving the underlying revisit-pressure issue intact.

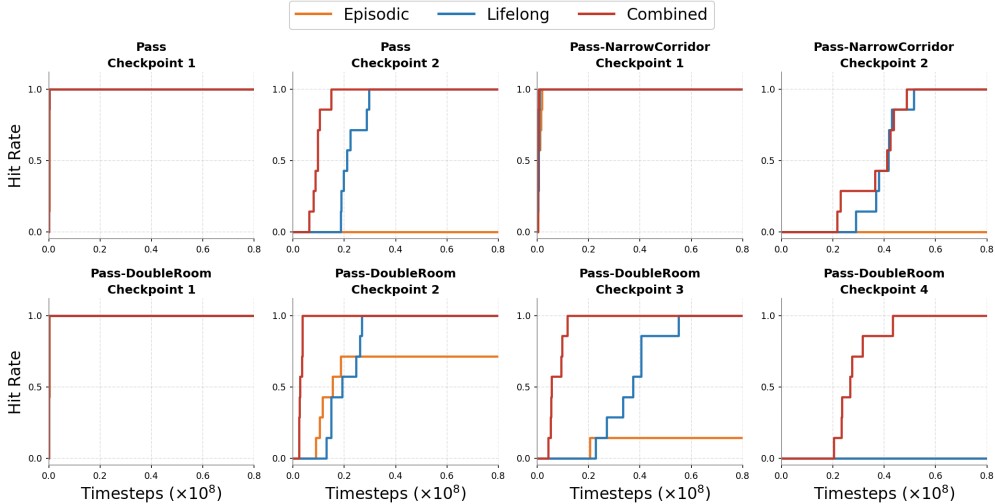

Figure 9: Cumulative checkpoint first-hit success rate across three environments for three exploration schemes, evaluated over 7 independent seeds. Each curve shows the fraction of seeds (out of 7) that have successfully hit a given checkpoint for the first time by a given training timestep.

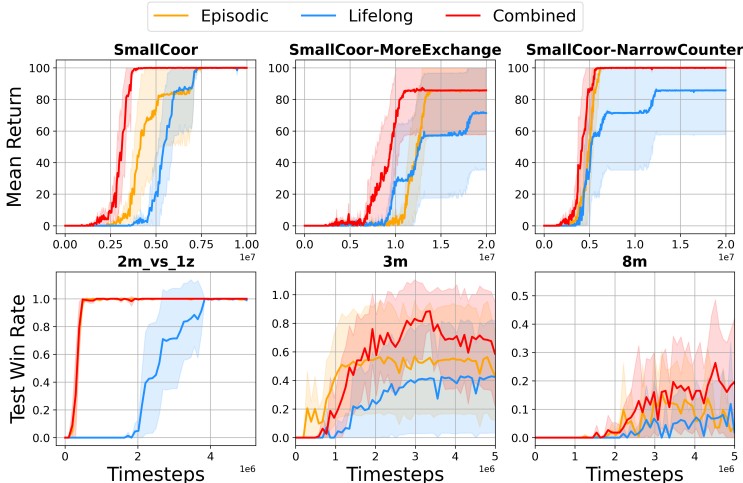

Figure 10: Learning curves of Episodic, Lifelong, and Combined Bonuses in Summation of Local Novelty in Overcooked and StarCraft tasks.

Taken together, these results demonstrate that our framework captures fundamental exploration–coordination trade-offs across discrete and continuous domains. Table 1 shows that in most environments and communication settings the combined approach achieves the highest or near-highest success rate. The main exception is the hardest Overcooked layout without communication (`SmallCoor-NarrowCounter`, $S = 1$), where no method learns reliably: the combined scheme remains at zero, lifelong-only also fails, and episodic-only only occasionally succeeds. Lifelong-only schemes are noticeably more sensitive to both geometry and communication and often require shared novelty information to reach good performance.

## 5.3 Alternative measures of episodic novelty

In the discrete-state setting (GridWorld), we use $\mathbb{I}(N_e^i(s^i) = 1)$ as the episodic novelty signal for agent $i$ in the main experiments (denoted "Episodic novelty 1"). We also tested an alternative count-based measure

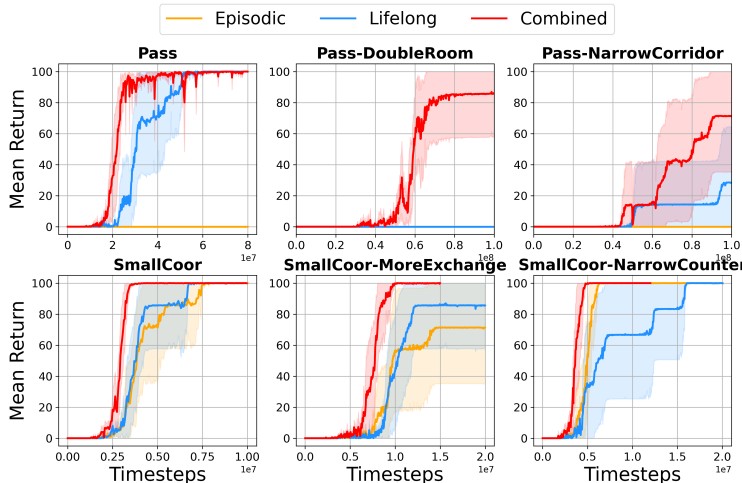

Figure 11: Learning curves of Episodic, Lifelong, and Combined Bonuses in MACE setting

| Environment | S | Episodic | Lifelong | Multiplicative Combined | Additive Combined |
|---|---|---|---|---|---|
| Pass | 1 | $0.00 \pm 0.00$ | $14.29 \pm 14.29$ | $\mathbf{100.00 \pm 0.00^*}$ | $0.00 \pm 0.00$ |
| | 2 | $0.00 \pm 0.00$ | $70.75 \pm 18.27$ | $\mathbf{100.00 \pm 0.00^*}$ | $0.00 \pm 0.00$ |
| | 3 | $0.00 \pm 0.00$ | $\mathbf{100.00 \pm 0.00}$ | $100.00 \pm 0.00$ | $0.00 \pm 0.00$ |
| Pass-NarrowCorridor | 1 | $0.00 \pm 0.00$ | $0.00 \pm 0.00$ | $\mathbf{99.98 \pm 0.02^*}$ | $0.00 \pm 0.00$ |
| | 2 | $0.00 \pm 0.00$ | $29.07 \pm 18.18$ | $\mathbf{100.00 \pm 0.00^*}$ | $0.00 \pm 0.00$ |
| | 3 | $0.00 \pm 0.00$ | $28.57 \pm 18.44$ | $\mathbf{72.98 \pm 17.48}$ | $0.00 \pm 0.00$ |
| SmallCoor | 1 | $29.81 \pm 18.11$ | $12.45 \pm 11.93$ | $\mathbf{100.00 \pm 0.00^*}$ | $43.29 \pm 20.06$ |
| | 2 | $\mathbf{100.00 \pm 0.00}$ | $\mathbf{100.00 \pm 0.00}$ | $\mathbf{100.00 \pm 0.00}$ | $\mathbf{100.00 \pm 0.00}$ |
| | 3 | $\mathbf{100.00 \pm 0.00}$ | $\mathbf{100.00 \pm 0.00}$ | $\mathbf{100.00 \pm 0.00}$ | $\mathbf{100.00 \pm 0.00}$ |
| SmallCoor-NarrowCounter | 1 | $\mathbf{14.29 \pm 14.29}$ | $0.00 \pm 0.00$ | $0.00 \pm 0.00$ | $0.00 \pm 0.00$ |
| | 2 | $\mathbf{100.00 \pm 0.00}$ | $85.66 \pm 14.28$ | $\mathbf{100.00 \pm 0.00}$ | $\mathbf{100.00 \pm 0.00}$ |
| | 3 | $\mathbf{100.00 \pm 0.00}$ | $\mathbf{100.00 \pm 0.00}$ | $\mathbf{100.00 \pm 0.00}$ | $\mathbf{100.00 \pm 0.00}$ |

Table 1: Mean Return (Mean±Std) over 7 seeds of Episodic, Lifelong, and Multiplicative Combined and Additive Combined Bonuses under three communication settings: $S = 1$ (Local Novelty only), $S = 2$ (Summation of Local Novelty), $S = 3$ (MACE). Bold entries highlight the best performance. The $*$ indicates statistically significant improvement for the best performance over the second-best performance using the Mann–Whitney U test ($p < 0.05$)

$1/\sqrt{N_e^i(s^i) + 1}$, matching the functional form used in our theoretical analysis (denoted "Episodic novelty 2") on `Pass`, `Pass-NarrowCorridor` and `Pass-DoubleRoom`.

In continuous-state settings (Overcooked and StarCraft II), we similarly compared two episodic metrics. The main experiments use a minimum-distance-to-episodic-memory term as described above ("Episodic novelty 1"). As an alternative ("Episodic novelty 2"), we evaluated an episodic curiosity signal inspired by Savinov et al. (2019) on `SmallCoor`, `SmallCoor-MoreExchange` and `2m_vs_1z`. As shown in Figure 12, all these variants yield the same qualitative conclusion: combining episodic and lifelong components consistently outperforms using lifelong novelty alone. This suggests that the benefits of the hybrid scheme are robust to the particular choice of episodic novelty metric.

## 5.4 Other algorithms

We also compare the three types of intrinsic motivation under a stronger policy-gradient backbone. Specifically, we adopt the optimism mechanism from OptiMAPPO (Zhao et al., 2024), a recent state-of-the-art multi-agent policy-gradient method. OptiMAPPO incorporates optimism into MAPPO by clipping advantages to remove negative values, which has been shown to accelerate multi-agent learning. To maintain the

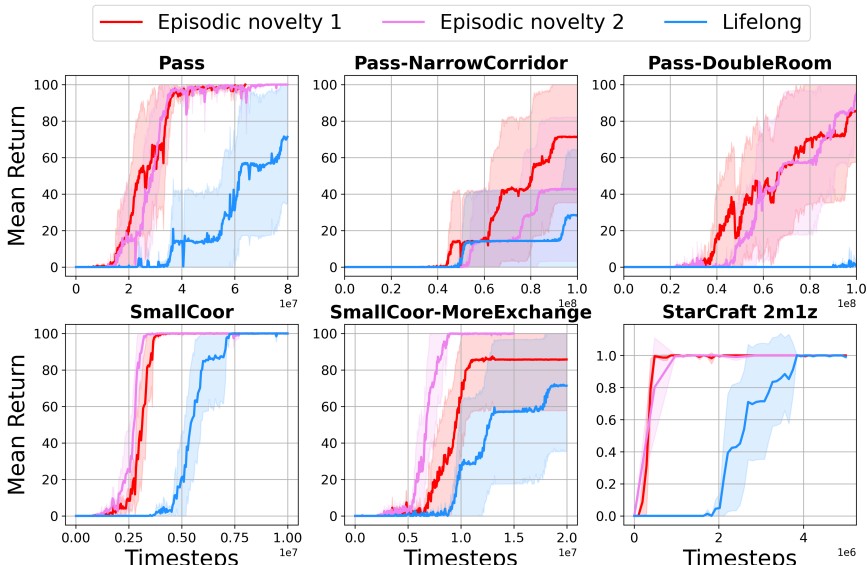

Figure 12: Different measures of episodic novelty. "Episodic novelty 1" refers to the metric used in the main experiments; "Episodic novelty 2" refers to alternative episodic signals in discrete and continuous settings. In all cases, the combined scheme outperforms lifelong-only exploration.

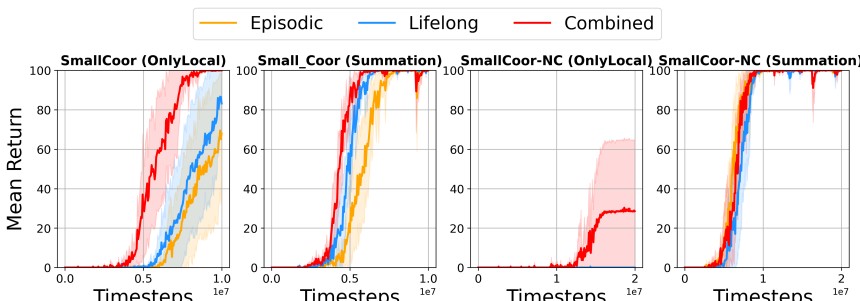

Figure 13: Comparison of three types of intrinsic motivation under an OptiMAPPO-style optimistic policy-gradient backbone. "OnlyLocal" denotes the local novelty setting; "Summation" denotes the summation-of-local-novelty setting. In all configurations, the combined episodic–lifelong scheme achieves the strongest performance.

decentralised setting, we adapt this optimism strategy within our IPPO-style implementation rather than relying on a centralised critic.

Figure 13 reports the results for the three intrinsic motivation schemes under this optimistic policy-gradient variant, in both the local novelty and summation-of-local-novelty communication settings. Even under this stronger algorithmic backbone, the hybrid bonus generally achieves the strongest performance, while the relative weaknesses of purely lifelong and purely episodic schemes mirror those observed with plain IPPO. This indicates that the advantages of the combined scheme are not specific to a particular base algorithm.

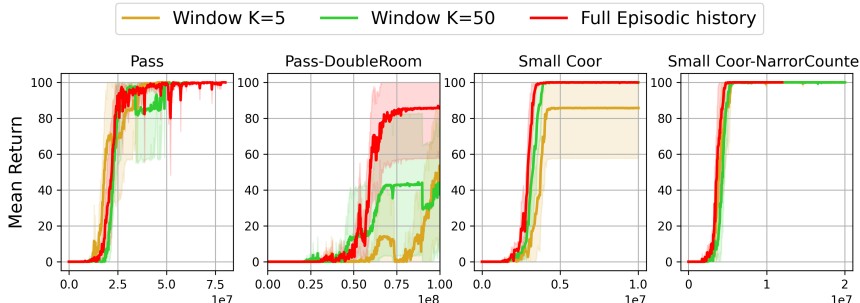

Figure 14: Ablation study of different episodic-memory window sizes when computing the episodic novelty term. Longer episodic memory yields larger gains in complex sequential tasks such as `Pass-DoubleRoom`.

### 5.5    Ablation study

To probe the mechanism of the episodic bonus, we conducted an ablation study on the size of its memory window (Figure 14). The results show that in complex sequential tasks such as `Pass-DoubleRoom`, using the full episodic history is critical for good performance, whereas a short episodic memory offers substantially less benefit. This task-dependent sensitivity supports our theoretical premise: for an episodic bonus to prevent novelty exhaustion at a key coordination point, its memory must be long enough to recognise that the location has already been visited *within the current episode*. In simpler tasks such as the Overcooked layouts, where the coordination loops are tighter and episodes shorter, a shorter memory is sufficient to yield gains. These results are consistent with the view that preventing coordination failure requires tracking within-episode visits at critical checkpoints.

### 5.6    Scalability

We also examine scalability with respect to team size in StarCraft II. In the `8m` scenario (eight Marines per side), the combined approach maintains its advantage over lifelong-only and episodic-only schemes (Fig. 10), confirming that the qualitative benefits of the hybrid bonus persist at larger team sizes in fully decentralised settings.

### 5.7    Per-episode revisit cap

Figure 15 compares lifelong-only, episodic-only, and combined novelty bonuses in terms of exploration behaviour in `Pass` under the local-novelty-only setting. The maximum episode length is $H = 301$, and we record the states visited by one agent within each episode during training.

Figure 15a shows that the combined method achieves the highest ratio of unique states visited per episode, indicating broader within-episode exploration. Figure 15b shows the maximum per-episode state revisit count, $\max_s N_e(s)$. Lifelong-only exploration often revisits the same states excessively, whereas the episodic-only and combined schemes maintain substantially lower peak revisit counts. This empirical pattern supports the effective per-episode revisit cap $C_{\text{epi}} \ll H$ used in the theory. Figure 15c shows cumulative unique states visited over training. The combined scheme initially discovers new states fastest, but later plateaus once it finds a viable route to complete the task. Lifelong-only exploration continues to visit new states less effectively because it fails to stabilise the coordinated behaviour required to reach later regions, while episodic-only exploration lacks persistent cross-episode directionality.

## 6    Discussion and conclusion

We have identified coordination de-synchronisation as a key failure mode in decentralised multi-agent exploration and developed a stylised theoretical framework for analysing when different intrinsic motivation schemes are favourable. Within this framework, exploration performance is governed by two structural quan-

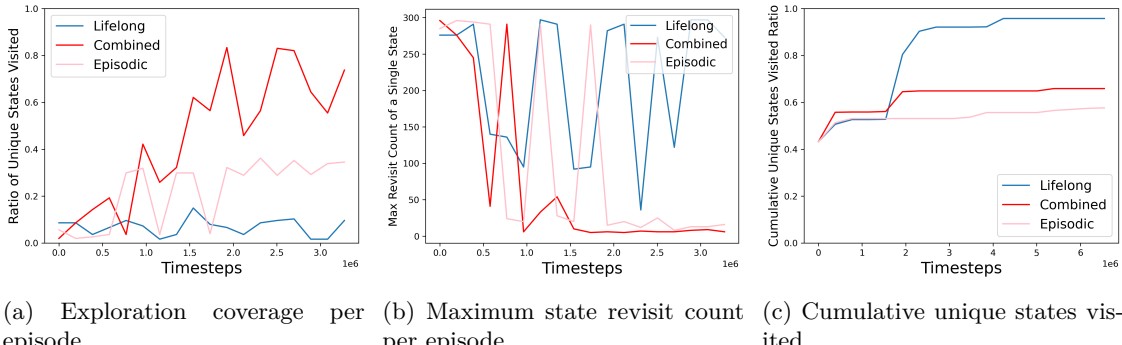

(a) Exploration coverage per episode

(b) Maximum state revisit count per episode

(c) Cumulative unique states visited

Figure 15: Exploration diagnostics for lifelong-only, episodic-only, and combined bonuses in `Pass` under local novelty only. Left: unique-state coverage per episode. Middle: maximum revisit count of any state within an episode. Right: cumulative unique states visited over training. Episodic-only and combined bonuses reduce repeated within-episode revisits relative to lifelong-only exploration. The combined bonus additionally maintains stronger long-term discovery because the lifelong component preserves cross-episode directionality.

tities: the coordination complexity $L$ (the number of sequential checkpoints that must be visited jointly) and the geometric revisit pressure $\beta$ (how strongly the environment forces repeated traversal of early checkpoints). Our analysis shows that, under mild behavioural assumptions, the guaranteed success probability under lifelong bonuses deteriorates at least polynomially with episode index at a rate controlled by $\beta N/2$, while episodic bonuses provide a time-uniform lower bound on re-coordination probability within the stylised model, but do not, on their own, bias exploration toward unseen regions across episodes. A hybrid scheme that multiplicatively combines episodic and lifelong novelty resolves this tension: it inherits a finite-window re-coordination lower bound from the episodic component, preserves an outward exploratory drive from the lifelong component, and yields $\beta$-independent finite-window guarantees for reaching deep sequences of checkpoints under the stated assumptions.

Experiments in GridWorld, Overcooked, and StarCraft II support these qualitative predictions. By varying $L$ and $\beta$ explicitly in GridWorld and implicitly through layout geometry and emergent strategies in Overcooked and StarCraft II, we observe the patterns suggested by the theory: (i) lifelong-only exploration performs adequately in low-complexity, low-pressure settings such as `Maze`, but degrades sharply as $L$ and $\beta$ increase; (ii) episodic-only exploration maintains re-coordination ability but struggles to discover and stabilise later checkpoints; and (iii) the hybrid scheme is consistently robust, particularly in environments with multiple sequential coordination requirements or narrow bottlenecks. The communication experiments further show that adding information sharing mostly affects constants in our bounds (through $\lambda_i$ and $\rho_{\min}$) rather than the qualitative dependence on $L$ and $\beta$: communication helps, but does not remove the fundamental vulnerability of lifelong-only bonuses in high-pressure regimes.

Beyond validating the theory, these results provide practical guidance for designing exploration mechanisms in cooperative MARL. In tasks with a single coordination event and relatively low revisit pressure (for example, one-shot rendezvous or simple room-entry problems), standard lifelong bonuses may be sufficient. However, in tasks that require agents to pass through the same coordination points repeatedly whilst searching for later objectives, or in maps with tight corridors and bottlenecks, our analysis and experiments both suggest that episodic components become important: they slow the depletion of earlier checkpoints and preserve re-coordination incentives over the finite training window. The multiplicative hybrid scheme we study offers a simple, implementation-agnostic way to achieve this, and can be combined with a variety of base learners (IPPO, optimistic policy gradients, CTDE methods) and novelty estimators (counts, RND, episodic curiosity) with similar qualitative benefits.

Our work has several limitations that delimit the regime of validity of these conclusions. First, the theoretical results are derived under a set of stylised assumptions (A1–A7) that abstract away many algorithmic details, and the finite-time guarantee for lifelong exploration applies only to the first checkpoint. The behaviour

of lifelong schemes beyond this first checkpoint, and under more complex function approximation, remains an open analytical question. Secondly, the geometric pressure parameter $\beta$ is estimated empirically rather than derived from first principles, and our guarantees are sufficient conditions rather than tight characterisations of performance. Thirdly, our theoretical framework focuses on fully decentralised settings with finite, discrete action spaces and local observations that uniquely determine each agent's local state; extending the analysis to continuous action spaces, more general observation models, and explicitly modelled communication protocols is an important direction for future work. The theoretical guarantees should therefore be interpreted as sufficient-condition results for sequential coordination problems with bottlenecks, not as universal guarantees for arbitrary MARL environments. The assumptions are most directly matched by the controlled GridWorld tasks. In Overcooked and StarCraft II, we use the framework to explain analogous empirical patterns involving repeated handoffs, bottleneck regions, and recurrent tactical configurations.

Despite these caveats, we believe the main conceptual message is robust: in decentralised multi-agent systems, the temporal scope of intrinsic motivation (lifelong versus episodic) interacts strongly with the structure of coordination requirements and environmental geometry. Treating episodic bonuses not merely as a single-agent trick for avoiding detachment, but as a mechanism for preserving re-coordination ability in the presence of lifelong exploration, appears to be a useful design principle. Future work could investigate adaptive schemes that modulate the balance between episodic and lifelong components in response to online signals of coordination failure, extend the analysis to richer forms of intrinsic motivation (for example, empowerment or information gain), and explore applications in real-world domains where coordination bottlenecks and revisitation are inherent, such as multi-robot systems, traffic management, or distributed sensor networks.

**Acknowledgments** TZ acknowledges support from the UK Engineering and Physical Sciences Research Council (EPSRC EP/W523793/1), through the Statistics Centre for Doctoral Training at the University of Warwick. GM acknowledges support from a UKRI AI Turing Acceleration Fellowship (EPSRC EP/V024868/1).

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
