# A    Formal assumptions

In this appendix we state the formal versions of Assumptions A1–A7 referenced in Section 3.4. They are intended as a stylised model of the agents' behaviour under intrinsic motivation, rather than exact properties of any particular algorithm.

**Assumption 1 (Fixed policy snapshot (A1))** *Throughout the theoretical analysis we consider a frozen snapshot of training in which the joint policy $\pi = (\pi_1, \ldots, \pi_N)$ is stationary across episodes. All expectations and visit-count bounds below are taken with respect to the Markov chain induced by this fixed policy. When we refer to a window of $K_{\max}$ episodes, we mean any contiguous block of $K_{\max}$ episodes under this stationary policy.*

**Assumption 2 (Exploration–mixing (A2))** *For each agent $i$, let $\mathcal{S}^i$ denote its finite local state space and let $\mathcal{S}^i_{\mathrm{reach}}(k) \subseteq \mathcal{S}^i$ denote the set of local states reachable from the current start distribution in episode $k$. Define*

$$\bar{r}^i_k(s) = \frac{r^i_k(s)}{\sum_{u \in \mathcal{S}^i} r^i_k(u)}.$$

*There exist multiplicative comparison constants $0 < \lambda_i \leq 1 \leq \kappa_i < \infty$ such that, for every episode $k$ prior to task completion and every $s \in \mathcal{S}^i_{\mathrm{reach}}(k)$,*

$$\lambda_i \, \bar{r}^i_k(s) \leq d^i_k(s) \leq \kappa_i \, \bar{r}^i_k(s),$$

*where $r^i_k$ is the intrinsic bonus at the beginning of episode $k$, and $d^i_k(s)$ is the probability that agent $i$ visits $s$ at least once during that episode. No assumption is made about states outside $\mathcal{S}^i_{\mathrm{reach}}(k)$.*

**Assumption 3 (Bounded cross-agent correlation (A3))** *There exist constants $0 < \rho_{\min} \leq 1 \leq \rho_{\max} < \infty$ such that, for every joint state $s = (s^1, \ldots, s^N)$ and episode $k$,*

$$\rho_{\min} \prod_{i=1}^{N} d^i_k(s^i) \leq \Pr\big(\exists\, t \leq H : s_t = s \text{ in episode } k\big) \leq \rho_{\max} \prod_{i=1}^{N} d^i_k(s^i),$$

*where $s_t$ denotes the joint state at time $t$ within the episode, and $d^i_k(s^i)$ denotes the probability that agent $i$ visits $s^i$ at least once during episode $k$.*

**Assumption 4 (Revisit-rate exponent (A4))** *For each agent $i$ there exist constants $c_i > 0$ and $\beta_i \geq 0$ such that for its first-checkpoint component $s^*_{i,1}$ and every episode $k$ before task completion,*

$$N^i_k(s^*_{i,1}) \leq c_i \, k^{\beta_i}.$$

*We define $\beta := \max_i \beta_i$ as the geometric revisit pressure.*

**Assumption 5 (Frontier exploration (A5))** *If a state $s$ has never been visited by agent $i$ up to episode $k$ then $N^i_k(s) = 0$ and hence $r^i_k(s)$ attains its maximal value.*

**Assumption 6 (Gated reachability (A6))** *The ordered checkpoint sequence is feasible within the episode horizon in a staged sense. There exist positive integer stage budgets $h_1, \ldots, h_L$ with*

$$\sum_{\ell=1}^{L} h_\ell \leq H.$$

*Define $T_0 = 0$ and $T_\ell = \sum_{m=1}^{\ell} h_m$. For each episode $k$, let*

$$E_{\ell,k} = \left\{ \exists t \in \{T_{\ell-1} + 1, \ldots, T_\ell\} : s_t = s^*_{(\ell)} \right\}$$

*be the event that checkpoint $\ell$ is reached during its allocated stage.*

*There exist constants $q_1, \ldots, q_L \in (0, 1]$ such that, for each $\ell = 1, \ldots, L$,*

$$\Pr(E_{\ell,k} \mid E_{1,k} \cap \cdots \cap E_{\ell-1,k}) \geq q_\ell \, \rho_{\min} \prod_{i=1}^{N} d_k^i(s_{i,\ell}^*),$$

*where the conditioning event for $\ell = 1$ is interpreted as the whole probability space. We define*

$$q_{\text{gate}} := \min_{\ell=1,\ldots,L} q_\ell > 0.$$

*Thus, feasibility requires an executable staged path through the checkpoints within the horizon, rather than only a generic lower bound on the horizon length.*

**Assumption 7 (Per-episode revisit cap (A7))** *When an episodic component is present, every agent $i$ visits any state at most $C_{\text{epi}} \geq 1$ times within a single episode, i.e.*

$$\sum_{t=1}^{H} \mathbb{I}(s_t^i = s) \; \leq \; C_{\text{epi}} \quad a.s.$$

## B   Proofs

### B.1   Proof of Lemma 1

At the start of each episode $k$, the episodic visit counts $N_{e,k}^i(s)$ are reset to zero for all states $s$. By its definition, the episodic bonus $r_{\text{epi},k}^i(s) = [N_{e,k}^i(s) + 1]^{-1/2}$ is therefore equal to 1 for all states. The sum of bonuses for agent $i$ is thus $\sum_{s' \in \mathcal{S}^i} r_{\text{epi},k}^i(s') = |\mathcal{S}^i|$.

We then apply Assumption 2, which links bonuses to agent behaviour. The probability of visiting any state $s$ is lower-bounded by $d_k^i(s) \geq \lambda_i \frac{r_{\text{epi},k}^i(s)}{\sum_{s'} r_{\text{epi},k}^i(s')} = \lambda_i / |\mathcal{S}^i|$. Finally, using Assumption 3 to couple the individual agent probabilities, the joint probability of hitting checkpoint $s_{(1)}^*$ is $p_k^{\text{epi}} \geq \rho_{\min} \prod_{i=1}^{N} d_k^i(s_{i,1}^*) \geq \rho_{\min} \prod_{i=1}^{N} \frac{\lambda_i}{|\mathcal{S}^i|}$. The bound on the expected hitting time follows from this constant success probability.

### B.2   Proof of Lemma 2

**1. Lower bound on the bonus.** By Assumption 4, $N_k^i(s_{i,1}^*) \leq c_i \, k^{\beta_i}$. Hence

$$r_{\text{life},k}^i(s_{i,1}^*) \; = \; \left(N_k^i(s_{i,1}^*) + 1\right)^{-1/2} \; \geq \; (1 + c_i)^{-1/2} \, k^{-\beta_i/2}.$$

**2. Lower bound on the single–agent visitation probability.** Assumption 2 gives, for every agent $i$,

$$d_k^i(s_{i,1}^*) \; \geq \; \frac{\lambda_i}{|\mathcal{S}^i|} \, (1 + c_i)^{-1/2} \, k^{-\beta_i/2}.$$

**3. Joint synchronisation probability.** Using the lower half of Assumption 3,

$$
\begin{aligned}
p_k \; &= \; \Pr\big(s_{(1)}^* \text{ reached in episode } k\big) \\
&\geq \; \rho_{\min} \prod_{i=1}^{N} d_k^i(s_{i,1}^*) \\
&\geq \; \rho_{\min} \prod_{i=1}^{N} \frac{\lambda_i}{|\mathcal{S}^i|} \, (1 + c_i)^{-1/2} \, k^{-\beta_i/2} \\
&=: \; C_{\min} \, k^{-\sum_i \beta_i/2} \; \geq \; C_{\min} \, k^{-\beta N/2},
\end{aligned}
$$

where $\beta := \max_i \beta_i$ and $C_{\min} := \rho_{\min} \prod_{i=1}^{N} \frac{\lambda_i}{|\mathcal{S}^i|}(1 + c_i)^{-1/2}$.

**4. Tail probability of never succeeding.** Let $\tau_{\text{life}}$ be the first episode that hits $s^*_{(1)}$. Because the joint policy is frozen during the window (Assumption 1) and each episode is generated with fresh environment randomness, we have

$$\Pr(\tau_{\text{life}} > k) \;=\; \prod_{t=1}^{k}(1 - p_t) \;\leq\; \exp\Bigl(-\sum_{t=1}^{k} p_t\Bigr).$$

Set $\alpha := \beta N/2 < 1$. Using the bound from Step 3,

$$\sum_{t=1}^{k} p_t \;\geq\; C_{\min} \sum_{t=1}^{k} t^{-\alpha} \;\geq\; \frac{C_{\min}}{1-\alpha}\bigl(k^{1-\alpha} - 1\bigr),$$

so

$$\Pr(\tau_{\text{life}} > k) \;\leq\; \exp\Bigl(-\tfrac{C_{\min}}{1-\alpha}\, k^{1-\alpha}\Bigr).$$

**5. Finite first moment when $\beta N < 2$.** Finally,

$$\mathbb{E}[\tau_{\text{life}}] = \sum_{k \geq 0} \Pr(\tau_{\text{life}} > k) \;\leq\; 1 + \sum_{k \geq 1} \exp\Bigl(-\tfrac{C_{\min}}{1-\alpha}\, k^{1-\alpha}\Bigr) < \infty,$$

because the series in $k$ has sub-exponential decay for $0 < 1 - \alpha < 1$.

**Remark 1 (Tightness of the hitting-time bound)** *The bound above is used only to establish a clean sufficient condition for finite expected hitting time. It is not intended to be tight. A sharper expression could be obtained by retaining the finite summation structure in $\sum_{t=1}^{k} p_t$ and bounding the resulting exponential series more carefully. This would improve constants but would not change the qualitative dependence on $\beta N/2$, which is the quantity relevant to our comparison between lifelong and episodic/hybrid novelty.*

### B.3 Proof of Lemma 3

**(i) Inward Stability.** For a known state and any $k \leq K_{\max}$, we have $r^i_{\text{epi},k} \geq 1/\sqrt{H+1}$ and $r^i_{\text{life},k} \geq 1/\sqrt{K_{\max}H + 1}$. Hence

$$\eta_i = \frac{\lambda_i}{|\mathcal{S}^i|}\frac{1}{\sqrt{H+1}\sqrt{K_{\max}H+1}}, \quad p^{\text{hyb}}_k \geq \rho_{\min} \prod_{i=1}^{N} \eta_i \geq \rho_{\min}\,\eta^N.$$

**(ii) Outward Discovery.** For an unseen state $s \notin \mathcal{F}_k$, $N^i_k(s) = N^i_{e,k}(s) = 0$, so $r^i_{\text{hyb},k}(s) = 1$. Since every bonus is $\leq 1$, $\sum_{s'} r^i_{\text{hyb},k}(s') \leq |\mathcal{S}^i|$ and Assumption 2 gives $d^i_k(s) \geq \lambda_i/|\mathcal{S}^i|$.

### B.4 Proof of Lemma 4

Fix a frozen-policy analysis window consisting of episodes $k_0 + 1, \ldots, k_0 + K_{\max}$, each of horizon $H$. Let $q_{\text{gate}}$ be the staged-reachability constant from Assumption 6, and suppose

$$B_{\max} \geq \max_{i,\ell} N^i_{k_0}(s^*_{i,\ell}).$$

**(1) Lower bound on the hybrid bonus at checkpoint components.** Assumption 7 implies that, within the $K_{\max}$-episode window, every agent visits any given state at most $K_{\max}C_{\text{epi}}$ additional times. Therefore, for every agent $i$, every checkpoint component $s^*_{i,\ell}$, and every episode $k$ in the window,

$$N^i_k(s^*_{i,\ell}) \leq B_{\max} + K_{\max}C_{\text{epi}}.$$

Hence

$$r^i_{\text{life},k}(s^*_{i,\ell}) = \left(N^i_k(s^*_{i,\ell}) + 1\right)^{-1/2} \geq (B_{\max} + K_{\max}C_{\text{epi}} + 1)^{-1/2}.$$

At the beginning of each episode, episodic counts are reset, so the episodic multiplier at every local state is 1. Thus,

$$r^i_{\text{hyb},k}(s^*_{i,\ell}) \geq (B_{\max} + K_{\max}C_{\text{epi}} + 1)^{-1/2}.$$

**(2) From bonus to single-agent visitation probability.** Since all count-based bonuses are at most 1,

$$\sum_{s' \in \mathcal{S}^i} r^i_{\text{hyb},k}(s') \leq |\mathcal{S}^i|.$$

Applying Assumption 2 gives, for every checkpoint $\ell$,

$$d^i_k(s^*_{i,\ell}) \geq \frac{\lambda_i}{|\mathcal{S}^i|}(B_{\max} + K_{\max}C_{\text{epi}} + 1)^{-1/2}.$$

**(3) Joint synchronisation probability for a staged checkpoint.** Combining the single-agent bounds through Assumption 3 gives

$$\rho_{\min} \prod_{i=1}^N d^i_k(s^*_{i,\ell}) \geq \rho_{\min}(B_{\max} + K_{\max}C_{\text{epi}} + 1)^{-N/2} \prod_{i=1}^N \frac{\lambda_i}{|\mathcal{S}^i|} = \bar{q}^\dagger.$$

**(4) Induction over the ordered checkpoint stages.** By Assumption 6, conditional on checkpoints $1, \ldots, \ell - 1$ having been reached within their allocated prefix budget, checkpoint $\ell$ is reached during its allocated stage with probability at least

$$q_\ell \, \rho_{\min} \prod_{i=1}^N d^i_k(s^*_{i,\ell}) \geq q_{\text{gate}}\bar{q}^\dagger.$$

Therefore,

$$\Pr(E_{1,k} \cap \cdots \cap E_{L,k}) = \prod_{\ell=1}^L \Pr(E_{\ell,k} \mid E_{1,k} \cap \cdots \cap E_{\ell-1,k}) \geq (q_{\text{gate}}\bar{q}^\dagger)^L.$$

**(5) Finite-window and residual waiting-time bounds.** Let $G_k$ denote the event that episode $k$ completes all $L$ checkpoints in order. The previous step gives, for every episode in the window and every history before that episode,

$$\Pr(G_k \mid \text{history before episode } k) \geq (q_{\text{gate}}\bar{q}^\dagger)^L.$$

Hence, conditional on no success before the window,

$$\Pr\left(\tau^{\text{hyb}}_L > k_0 + K_{\max} \mid \tau^{\text{hyb}}_L > k_0\right) \leq \left(1 - (q_{\text{gate}}\bar{q}^\dagger)^L\right)^{K_{\max}},$$

which is equivalent to

$$\Pr\left(\tau^{\text{hyb}}_L \leq k_0 + K_{\max} \mid \tau^{\text{hyb}}_L > k_0\right) \geq 1 - \left(1 - (q_{\text{gate}}\bar{q}^\dagger)^L\right)^{K_{\max}}.$$

If the same lower bound is maintained after $k_0$ until the first successful episode, the residual hitting time is stochastically dominated by a geometric random variable with parameter $(q_{\text{gate}}\bar{q}^\dagger)^L$. Thus

$$\mathbb{E}\left[\tau^{\text{hyb}}_L - k_0 \mid \tau^{\text{hyb}}_L > k_0\right] \leq (q_{\text{gate}}\bar{q}^\dagger)^{-L}.$$

For $k_0 = 0$, this gives the stated expected hitting-time bound from the start of training.

## C  Environment Details

### C.1  GridWorld

The GridWorld environment is adapted from Jiang et al. (2024). The *door-switch* mechanism in `Pass` (Fig. 16) : Door 1 will open when switch 1 or switch 2 is occupied. Therefore, for both agents to reach the target room, the task includes two door-switch interactions: First, Agent 1 occupies the switch 1 and agent 2 enters the target room. Then, agent 2 occupies the switch 2 in the target room, and therefore agent 1 can also enter the target room. In GridWorld, each agent observes its own position $(x, y)$ along with the open/closed states of doors, represented as binary values ( 0 for closed and 1 for open). The action space includes four discrete actions, including *move up*, *move down*, *move left* and *move right*. Each episodes lasts up to 300 steps. The episode ends immediately once both agents reach the target room. Agents receive a global reward +100 only when both agents are in the target room.

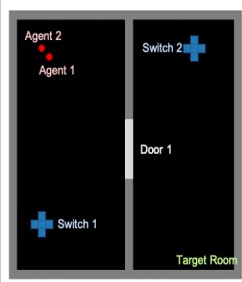

Figure 16: Gridworld:`Pass`

### C.2  Overcooked

Our tasks in Overcooked environment setting are also adapted from Jiang et al. (2024). Two agents must collaborate across an impassable kitchen counter. The left agent can access tomatoes, serving area or dishes (as in `SmallCoor-MoreExchange`), while the right agent has access to dishes and the pot. To complete a soup, one agent must place a tomato into the pot, cook it, serve it into a dish, and pass it through the counter. The environment has some modifications based on the open-source Overcooked environment Carroll et al. (2019).

- Restricted observation range. In the original setting, agents could observe all objects regardless of their distance. To better distinguish between global and local novelty, we introduce a configurable observation radius. Items outside this range are invisible to the agent. For instance, the left agent might be unable to see the pot or dishes on the right side.

- Instant soup cooking. To simplify the task, we remove the soup's cooking delay. Instead of the original 20-timestep wait, soup is cooked immediately upon interacting with the pot (i.e., with zero time cost).

- Episode termination on success. Rather than ending after a fixed number of steps, each episode concludes as soon as one correctly cooked soup is served. This focuses learning on the precise coordination required for a single successful completion, using one tomato per episode.

We use the featurized state representation, extended to include observation restrictions. The resulting observation vector has 38 dimensions. Each agent can only observe objects within its room or on the central counter. The agents operate in a discrete action space with six options: move up, move down, move left, move right, interact, and stay. The maximum episode length is set to 300 steps. Agents receives a global reward +100 only when finishing serving a soup.

### C.2.1 StarCraft Multi-Agent Challenges

The StarCraft Multi-Agent Challenge (SMAC) Samvelyan et al. (2019) is a widely adopted benchmark for cooperative multi-agent reinforcement learning. We use the open-source SMAC environment (MIT License) and evaluate performance on three representative maps: 2m_vs_1z, 3m, and 8m. Each episode lasts to 600 steps.

- In 2m_vs_1z, two Marines cooperate to defeat a single enemy Zealot. Agents receives +200 upon defeating the enemy.

- In 3m, three Marines must defeat three enemy Marines. Agents received +10 when defeating one enemy and received +200 when defeating all enemies.

- In 8m, eight Marines face off against eight enemy Marines. The sparse reward setting is the same as in 3m.

## D Explanation of different communication settings

We consider three communication settings of increasing complexity: local novelty only, summation of local novelty and MACE (Jiang et al., 2024). For local novelty only, there is no communication among agents and agents explore independently. For the summation of local novelty and MACE, agents are allowed to share a scalar value of local novelty mutually. Here, for the sake of consistency, we fixed the exchange novelty as lifelong novelty. Therefore, no matter what local novelty (Lifelong only, Episodic only and combined), agents always share lifelong novelty.

For MACE (Jiang et al., 2024), the intrinsic reward for each agent included three parts:

$$r_{int}^i(o_t^i, a_t^i) = \lambda_{self} * u_t^i + \lambda_{other} * \sum_{j \neq i} u_t^j + \lambda_h \sum_{j \neq i} v_{i,j}$$

where $u_t^i$ is the its own local novelty, $\sum_{j \neq i} u_t^j$ is the local novelties shared by other agents and $v_{i,j} = \sum_{j \neq i} z_t^j w_t^{i,j}$ is the term that quantify the influence of agent $i$ on the agent $j$, while $z_t^j = u_t^j + \gamma u_{t+1}^j + \cdots + \gamma^{T-t} u_T^j$ is the discounted accumulated novelties from time $t$ to the end of the episode of other agents, $w_t^{i,j} = \log \frac{p(a_t^i | o_t^i, z_t^i)}{\pi^i(a_t^i | o_t^i)}$ quantify the influence of agent $i$ on agent $j$'s future novelty. $\lambda_{self}, \lambda_{other}$ and $\lambda_h$ are coefficients for three components.

## E Empirical diagnostic of assumption A2 and A3

We empirically examine whether intrinsic novelty, implemented as a lifelong count-based bonus, is predictive of agents' exploration behaviour in a symmetric four-room environment under the local novelty-only setting. To eliminate positional confounds, the grid-world in Figure 17 is constructed such that all four rooms are equidistant from the central spawn point. At the beginning of each episode, we compute a **lifelong novelty estimate** for every grid cell based on the count-based intrinsic bonus. For each training epoch, we then estimate a **visitation frequency** for each grid cell, defined as the normalised number of visits divided by the total number of visits within the epoch. To quantify the relationship between novelty and exploration, we compute, for each epoch, the Pearson correlation

$$C_P = \text{Pearson}(\text{novelty}_{\text{start}}, \ \text{visitation}_{\text{epoch}}),$$

where each data point corresponds to a grid cell. Intuitively, this measures whether states with higher novelty at the start of an episode are more likely to be visited during training. Figure 18 shows the evolution of this correlation over epochs. In early training, the correlation is often low or negative. This is expected, as novelty is initially high and relatively uniform across the state space, while visitation is sparse and dominated by stochasticity, leading to weak or noisy alignment. As training progresses, the correlation steadily increases

and becomes predominantly positive. This indicates that states with higher novelty are increasingly more likely to be visited, suggesting that the intrinsic novelty signal becomes progressively more aligned with exploration behaviour. Occasional negative correlations can still occur when agents repeatedly visit a subset of states, causing their novelty to decrease while visitation remains high. Overall, these results are consistent with the hypothesis that intrinsic novelty provides a meaningful signal for guiding exploration and becomes increasingly predictive of visitation patterns as training proceeds.

We further evaluate Assumption 3 by comparing the empirical joint visitation probability with the product of marginal probabilities, as shown in Figure 19. For each epoch and every pair of rooms $(i, j)$, we compute the observed joint visitation probability $P_{\text{joint}}(i, j)$ and the product of marginals $P_0(i) \times P_1(j)$, where the marginals are derived from each agent's individual state distribution. Each scatter point corresponds to one room pair from one epoch. The points cluster tightly around the diagonal $y = x$, indicating that the joint visitation probability is well approximated by the product of the marginals in this diagnostic environment. This behaviour arises because the two agents explore the four equidistant rooms in a nearly independent manner. Consequently, Assumption 3 holds with multiplicative constants $\rho_{\text{min}}$ and $\rho_{\text{max}}$ close to 1, meaning that cross-agent correlation is bounded and weak in this setting.

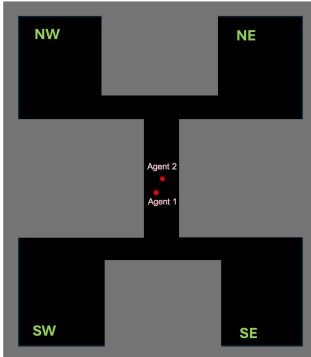

Figure 17: Four rooms with the same distances to the initial positions of agents; The four corner rooms (NW, NE, SW, SE) are equidistant from the agents' initial positions.

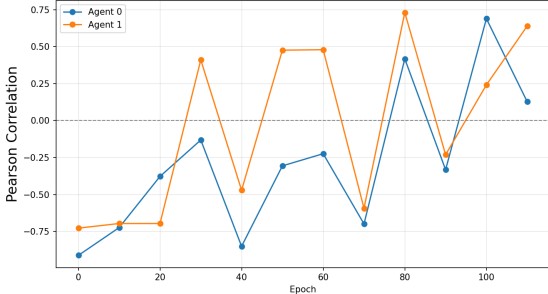

Figure 18: Pearson Correlations between Lifelong Novelty at the episode start and Visitation Probability within the episode

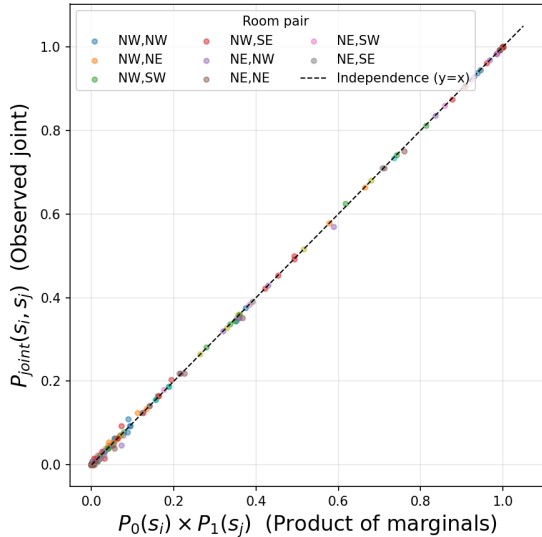

Figure 19: Joint visit probability and the product of marginal visit probability for two agents; Each scatter point represents one room pair $(s_i, s_j)$ at a given training epoch; all 16 room-pair combinations are plotted, with 8 representative pairs labeled in the legend.

## F Hyper-parameters

For the hyper-parameters of IPPO and the MACE component, we follow Jiang et al. (2024). Tables 2 and 3 report the intrinsic-reward coefficients used in each task. The episodic-only coefficient is much smaller than the coefficient used in the combined setting because the raw reward scales are different. Episodic-only novelty resets every episode and can remain persistently large, whereas in the combined setting the episodic signal is multiplied by the lifelong/RND term, which attenuates the realised intrinsic reward as training progresses. Figure 20 shows the coefficient-scaled realised intrinsic reward magnitudes, confirming that the effective scales are comparable after applying these coefficients. We ran all experiments on a server with NVIDIA TITAN V GPU (12GB, CUDA 11.4), Intel Xeon CPU, 128GB RAM, and Ubuntu 20.04. All neural networks were implemented using PyTorch.

Table 2: Specific hyper-parameters for Gridworld and Overcooked variants.

| Hyperparameter | Gridworld and Overcook | | |
|---|---|---|---|
| | Episodic-only | Lifelong-only | combined |
| $\lambda_{self}$ | 10.0 | 10.0 | 10.0 |
| $\lambda_{other}$ | 10.0 | 10.0 | 10.0 |
| episodic coef | 0.005 | - | 1.0 |

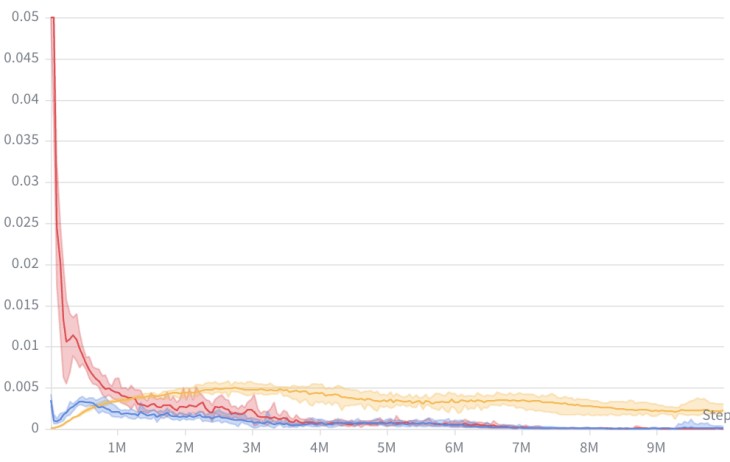

Figure 20: Coefficient-scaled realised intrinsic reward magnitudes over training in `SmallCoor`. Although the raw episodic-only and combined bonuses use different coefficients, the scaled intrinsic rewards have comparable magnitudes during training. This supports the coefficient choices in Tables 2 and 3.

Table 3: Specific hyperparameters for StarCraft.

| Hyperparameter | $2m\_vs\_1z$ | | | 3m/8m | | |
|---|---|---|---|---|---|---|
| | Episodic-only | Lifelong-only | Combined | Episodic-only | Lifelong-only | Combined |
| $\lambda_{self}$ | 0.2 | 0.2 | 0.2 | 0.2 | 0.2 | 0.2 |
| $\lambda_{other}$ | 0.2 | 0.2 | 0.2 | 0.2 | 0.2 | 0.2 |
| episodic coef | 0.005 | – | 1.0 | 0.005 | – | 2.0 |