# OpenReview forum: "When Lifelong Novelty Fails: Coordination Breakdown in Decentralised MARL"
_TMLR — Accepted by TMLR_

### Review · Reviewer_TBbT · 2026-01-02

**Summary Of Contributions:**

The paper studies how to assign novelty-based bonus in decentralized MARL, especially in environments with multiple key bottleneck joint coordination checkpoints. The authors argue that lifelong novelty bonus fails in such environments and introduce an alternative bonus that combines lifetime and episodic bonuses. Within a stylized analytical framework and under a number of assumptions, the authors derive bounds suggesting the decaying success rate under lifelong bonus with respect to the number of agents and a “revisit pressure” parameter, whereas they do not suffer this decay under the proposed bonus.

The theory is supported by experiments in GridWorld, Overcooked, and StarCraft II, across communication regimes, showing consistent degradation of lifelong-only exploration and robust performance of the hybrid approach in environments with high revisit pressure or coordination complexity.

On the negative side, I am concerned that there is a big gap between the proposed exploration bonus and the accompanying theoretical analysis: the theory relies on a number of strong and unclear assumptions, and it is very unclear how the assumptions relate to/are guaranteed by the actual proposed method.

**Audience:**

Yes

**Audience Explanation:**

How to systematically add exploration bonus in (multi-agent) RL seems an important subject to study. While I do not work on this subject, I believe there are researchers in the RL community who will find this work interesting.

**Broader Impact Concerns:**

No concerns

**Claims And Evidence:**

Yes

**Claims Explanation:**

The paper's claims are supported by a few theoretical results and a comprehensive set of experiments.

**Requested Changes:**

1) >Under partial observability, each agent computes novelty based only on its local observations, which may diverge significantly from the true novelty of the joint state.

I think this statement exaggerates the technical challenge. Any environment-defined rewards in MARL are also only locally accessible like this novelty-based reward.

2) Regarding the formulation in Section 3.1 -- The introduce of POMDP seems unnecessary, since the authors only look at the class of problems where local observation can unique determine the local state. It is better to just take the standard MARL formulation for clarity.

3) Why do the authors focus exclusively on the cooperative setting? What is particular about cooperation that makes the proposed analysis inapplicable, or less relevant, to mixed or competitive multi-agent environments?

4) Regarding Eq.(1) -- What does "next" mean in a global sense? I understand something like p_{k,i}=Pr(the first i checkpoints are visited before episode k), but the current definition is unclear.

5) If lifelong bonus becomes small, the hybrid bonus also does. This cannot be saved by the episodic return, right? But the experiments clearly show that the policy quality under hybrid bonus and lifelong bonus is very different, which I do not quite understand.

6) In my view, assumptions are among the most crucial aspects that any theoretically focused paper should clearly present. If short in space, the authors can consider moving some proof sketch to the appendix. The assumptions should appear in the main paper.

7) **Important.** Many assumptions (at least Assumption 2, 3, 4, 6, 7) are unclear and feel very strong. They also do not feel immediately connected to the proposed exploration bonus. Under what circumstances can these assumption be guaranteed to hold? The theoretical results are meaningless if the assumptions are not verifiable.

8) The authors argue that the revisit pressure is a key quantity characterizing the problem difficulty. This is introduced in Assumption 4 in the appendix but not in the main paper. Also, I do not quite see how to relate $\beta$ to something that fundamentally describes the problem difficulty.

---

> ### Author Response · Authors · 2026-04-24
>
> We thank the reviewer for the careful reading and for highlighting that the theoretical assumptions needed to be more visible, interpretable, and directly connected to the proposed exploration mechanism. We substantially revised the theoretical presentation in response.
>
> ### 1. Partial observability and local novelty
>
> We agree that our original wording overstated the issue. The challenge is not that rewards are locally accessible per se; environment rewards in decentralised MARL may also be observed locally or partially. The specific issue for novelty bonuses is that novelty is an agent-dependent statistic computed from local observations or local states, and this local novelty may not coincide with novelty of the joint state that matters for coordination.
>
> We revised the wording to avoid presenting this as a limitation unique to intrinsic rewards. The manuscript now states more precisely that local novelty can misalign with joint coordination novelty, especially when agents must synchronise at joint checkpoints that no individual agent can fully observe.
>
> ### 2. Section 3.1 formulation
>
> We agree that introducing the POMDP formalism was unnecessary for the theoretical class we analyse. We revised Section 3.1 to use a standard MARL formulation directly in terms of local states. We also clarify that the analysis focuses on the common subclass where each agent’s local observation uniquely determines its local state, while the global state remains partially observable because agents do not observe the other agents’ local states.
>
> ### 3. Why the analysis focuses on cooperative settings
>
> We focus on fully cooperative MARL because the mechanism analysed in the paper relies on all agents sharing the same task objective. Under a shared reward, intrinsic bonuses bias the agents’ visitation distributions toward states that can jointly support task progress, making assumptions such as exploration-mixing and bounded cross-agent correlation meaningful.
>
> In mixed or competitive settings, a state with high novelty for one agent may be strategically harmful, exploitable, or irrelevant to the other agents. Thus, novelty need not induce a monotone improvement in joint coordination probability, and the product-style lower bounds used in our analysis may fail. We added this scope clarification to the manuscript and now state that extending the analysis to mixed or competitive games would require different assumptions about strategic interaction and opponent response.
>
> ### 4. Meaning of “next” checkpoint in Equation 1
>
> We clarified the definition of $p_k$. The “next” checkpoint means the next uncompleted checkpoint in the ordered task sequence. More formally, if the highest completed checkpoint is $\ell-1$, then $p_k$ denotes the probability that the agents reach checkpoint $\ell$ during episode $k$. The expected hitting time $\tau_L$ then measures the number of episodes required to complete the full sequence of $L$ checkpoints. We revised Section 3.2 to make this interpretation explicit.
>
> ### 5. Why the hybrid bonus can outperform lifelong novelty even though the lifelong factor decays
>
> The reviewer is correct that a multiplicative hybrid bonus cannot restore a large absolute reward once the lifelong term has been fully depleted. We revised the manuscript to avoid implying an asymptotic non-decaying reward. The role of the episodic component is instead preventive and trajectory-shaping.
>
> Under lifelong-only novelty, agents may repeatedly cycle through bottleneck states within an episode, rapidly increasing lifelong visit counts at task-critical states and depleting their future bonus. The episodic factor suppresses repeated within-episode revisits, which changes the induced trajectory distribution and reduces excessive depletion of these critical states. Therefore, over a finite training window, the hybrid method can maintain a stronger usable signal at coordination checkpoints than lifelong-only exploration. This is also why the theoretical hybrid bound depends on the episodic revisit cap $C_{\mathrm{epi}}$, not on the episode horizon $H$.
>
> We added this clarification and now describe the hybrid guarantee as a finite-window, assumption-dependent coordination lower bound rather than an unconditional asymptotic floor.

---

> ### Author Response · Authors · 2026-04-24
>
> ### 6. Moving assumptions into the main paper
>
> We agree that the assumptions are central to the paper. We moved the assumptions into the main body as a dedicated subsection, with concise explanations of their role and scope, while keeping the formal versions and proofs in the appendix. This makes clear before the lemmas what each assumption means and why it is needed.
>
> ### 7. Strength, verifiability, and practical meaning of Assumptions A2, A3, A4, A6, and A7
>
> We revised the theory section to make the assumptions more interpretable and to state their scope as sufficient conditions rather than universal guarantees.
>
> A2 formalises the idea that, among reachable states, higher intrinsic reward should increase visitation probability. This is the behavioural link between the proposed bonus and the induced policy. A3 rules out degenerate cases where agents’ local visitation probabilities are completely uninformative about joint synchronisation. A4 defines the revisit-pressure exponent $\beta$, which measures how quickly bottleneck or checkpoint visit counts grow with training. A6 is a solvability/gated-reachability condition for sequential coordination tasks. A7 captures the effect of episodic memory in limiting excessive within-episode revisits.
>
> In the current revision, we provide empirical support for A4 and A7 and clarify the scope of A6. For A2 and A3, we now explicitly state that these are assumptions about behavioral regularities rather than algorithmic guarantees, and we discuss how they can be empirically checked in tabular domains—specifically, by comparing bonus-ranked visitation probabilities against each other and by comparing joint-hit probabilities to products of marginal hit probabilities. We have included a diagnostics for A2 and A3 in a simple GridWorld and local novelty-only exploration setting in the appendix. These diagnostics demonstrate that, in this controlled tabular environment, a higher intrinsic bonus is associated with a higher visitation probability and empirical joint visit probabilities are broadly comparable to products of marginal visitation probabilities.
>
> ### 8. Revisit pressure $\beta$ as a central difficulty quantity
>
> We expanded the main text discussion of $\beta$. Revisit pressure is defined as the exponent in the empirical growth law $N_k \approx C k^\beta$, where $N_k$ is the cumulative visit count to a critical bottleneck or checkpoint region. This quantity is directly relevant to lifelong novelty because count-based lifelong bonuses decay as $N_k^{-1/2}$. Therefore, if $N_k$ grows as $k^\beta$, the lifelong bonus at that state decays as $k^{-\beta/2}$, and the joint coordination lower bound for $N$ agents decays as $k^{-\beta N/2}$.
>
> Thus, $\beta$ is not intended to describe all forms of task difficulty. It isolates the geometric component of difficulty that specifically harms lifelong novelty: repeated traversal of task-critical states. We added this definition and interpretation to the main paper, together with empirical estimates of $\beta$ in the GridWorld layouts.

---

> > ### Comment · Reviewer_TBbT · 2026-04-26
> >
> > I appreciate the authors' detailed clarification, which addressed most of my original questions and remarks. The exception is item 7; I still find a subset of the assumptions difficult to make sense of. Upon realizing that the novelty reward itself is not an innovation of this work, as pointed out by reviewer RoFu, and that the main contribution should be understood as providing a theoretical analysis of novelty rewards, I feel that the restrictiveness and interpretability of these assumptions undermines the overall contribution.
> >
> > That said, I understand that the problem itself is likely not well structured for direct mathematical analysis, and as an initial step toward a more complete understanding, it is reasonable to begin with a set of simplifying assumptions. Overall, I find the paper well written, the presentation clear, and the core message solid and reasonable. I therefore (weakly) support the acceptance of the paper.

---

### Review · Reviewer_whWG · 2026-01-23

**Summary Of Contributions:**

Exploration in Reinforcement Learning (RL) relies on lifelong novelty bonuses, exploration-based rewards awarded to agents for exploring novelty states in an environment. The paper identifies coordination desynchronisation in Multi-Agent Reinforcement Learning (MARL), a coordination-exploration tradeoff phenomenon. Agents tend to repeateadly traverse previously explored checkpoints when searching for novel coordination checkpoints. This leads to a degradation in intrinsic motivation, and hence, long-term exploration capabilities of agent populations requiring coordination. The paper studies coordination desynchronisation by theoretically evaluating it in settings of episodic, lifelong and hybrid (episodic and lifelong) novelty bonuses over varying level of environment coordination complexity and checkpoint revisit pressure. Theoretically, authors present lower bounds on probability of success and upper bounds on expected checkpoint hitting times. Empirically, experiments demonstrate novelty varies with revisit pressure and coordination complexity. Hybrid novelty bonuses are found to be aligned with theoretical claims and improve coordination synchronisation across studied domains.

**Additional Comments:**

NA

**Audience:**

Yes

**Audience Explanation:**

Coordination in MARL is an important concept requiring attention both from a theoretical and empirical standpoint. The paper studies exploration across varying levels of coordination in MARL. Empirical results support theoretical claims and aim to deonstrate the significance of both episodic and lifelong novelty bonusses in MARL exploration. The work will be of interest to members of the RL and MARL communities.

**Claims And Evidence:**

No

**Claims Explanation:**

* **Sequential Checkpoint Dependency:** The paper considers tasks where checkpoints form a one dimensional markov chain, i.e- completion of one checkpoint leads to the next checkpoint. Assumptions and practical considerations further support this setup in isolating exploration towards coordination-based behaviors. However, one aspect remains which may indirectly influence exploration. To the best of my understanding, agent roles are not prefixed. For instance, Agent 1 and Agent 2 both may choose to activate _switch_ or _open door_. Could the authors explain how roles and heterogenous agent compositions alter exploration? Are agent roles decided apriori or learned during episodic interations? Since agents can only explore from a set of states which are allowed corresponding to their roles, it remains unclear as to how these sets are constructed and upto what extent they influence coordination novelty. Could the authors also explain the case wherein one checkpoint can be attained from multiple prior checkpoints? As per the markov chain assumption, the previous checkpoint encapsulates attainment of prior checkpoints. However, in large state spaces with multiple checkpoints, there exist combinatorial ways to reach a particular coordination checkpoint, eg- multiple possible doors leading to the next door. Furthermore, it is unclear whether agents prioritize a set of states (as per their roles) or they explore randomly? This would provide insights into whether exploration is role-agnostic / agent-agnostic.

* **Empirical Evaluation:** While experiments and empirical analysis is comprehensive, a few results could be better put into perspective using the episodic novelty bonuses. Qualitatively, it is unclear as to how do episodic bonuses impact behaviors in Gridworld. It also remains unclear how revisits vary for episodic bonuses. Since episodic memory dominates in short-horizon tasks consisting of multiple agents due to initialization and tracking visitations from scratch, authors should add these comparisons. Furthermore, the impact of revisit pressure $\beta$ and coordination complexity $L$ could be made more clear. Explicitly observing the variation of returns against $\beta$ and $L$ would make comparisons clear. Finally, the work can better evaluate novel uncovered states and behaviors. While Figure 13 is comprehensive, variations can also be studied using the cumulative unique states for both hybrid and episodic bonuses. These can also be evaluated using the rate of hitting new checkpoints/states as a faster rate indicates both rapid coordination and exploration. Such comparisons will comprehensively indicate whether hybrid bonuses actually discover new states during cooperation at a faster rate while making use of both lifelong and episodic components.

### Minors

* Equation 2: can the definition be made more intuitive or explained in text? eg- number of episodes required to hit checkpoint L
* Lemma 1: What are $\lambda_{i}$ and $\mathcal{S}^{i}$?
* Lemma 2: What is $c_{i}$?
* Figure 1 (c): can you please explain what is the third coordination checkpoint?
* Appendix B.2 Step 5: Following the splitting of summation into $k = 0$ and $k \geq 1$, expectation is upper bounded with $\infty$. From an empirical point, it establishes finiteness and is trivial. From a theoretical standpoint, can the bound be tightened (given that it is a sum of exponentials)?

**Requested Changes:**

Following are the changes critical for acceptance -

* Explanation of the impact of roles and agent compositions (homogenous or heterogenous agent populations) and how they impact exploration in the overall setup. It would also be worthwhile to explain the checkpoint reachability from one or many prior checkpoints for a set of agents.
* Complete empirical analysis consisting of episodic bonus results (in Figure 9 and Figure 13).
* Evaluation of novel uncovered states. This can be achieved by studying variation of cumulative unique states and the rate of hitting new checkpoints/states for episodic and hybrid bonuses.

Following are the changes that would strengthen the quality of work -

* Explicit variation of coordination performance with $\beta$ and $L$.
* Tighter bound on expected hitting time given that the expression simplifies to a sum of exponentials.
* Minor errors in text.

---

> ### Author Response · Authors · 2026-04-24
>
> We thank the reviewer for the detailed comments and for identifying several places where the role of agent identities, checkpoint structure, and episodic-only comparisons needed to be made clearer. We revised both the theoretical discussion and the empirical analysis accordingly.
> ### 1. Role assignment, homogeneous agents, and heterogeneous agents
>
> Agent roles are not predefined in our experiments. Agents are indexed for learning and notation, but their behavioural roles emerge through interaction with the environment and the intrinsic/extrinsic rewards. In homogeneous settings, all agents share the same action capabilities, so either agent may in principle activate a switch, traverse a door, or perform a complementary coordination action. The notation $s^{*}_{i,\ell}$ indexes the local component of the joint checkpoint for agent $i$; it should not be interpreted as a manually assigned role.
>
> We clarified this in the revised manuscript. For homogeneous agents, when several role assignments are valid, a checkpoint can equivalently be represented as a set of valid joint states, including permutations of the agents’ physical roles. Our analysis applies to any such feasible assignment; allowing multiple feasible assignments only increases the probability of reaching the checkpoint. For heterogeneous agents, the feasible local state/action spaces are agent-specific. This is already reflected in the theory through agent-specific quantities such as $\mathcal{S}^i$, $\lambda_i$, $c_i$, and $\beta_i$. Heterogeneity therefore changes the constants and the set of reachable joint checkpoints, but does not require roles to be fixed manually.
>
> ### 2. Multiple prior checkpoints and non-linear reachability
>
> We agree that real environments need not form a single one-dimensional checkpoint chain. The chain structure in the theory is a stylised abstraction used to isolate the sequential coordination failure mode. In environments with multiple doors, alternative handoff points, or several possible predecessor checkpoints, the natural structure is a checkpoint graph rather than a single chain.
>
> We revised the manuscript to clarify that the theoretical chain should be interpreted as one feasible path through this graph, or as a dominant successful trajectory induced by learning. Multiple possible predecessors do not invalidate the mechanism; they can reduce effective difficulty by increasing the number of ways to reach a later checkpoint. However, once progress repeatedly requires returning through earlier bottlenecks or coordination states, the same lifelong novelty depletion mechanism applies along any selected path. Agents do not prioritise a manually specified set of role-dependent states; their trajectories are induced by the learned decentralised policy shaped by intrinsic and extrinsic rewards.
>
> ### 3. Episodic-only comparisons in the empirical analysis
>
> We agree that episodic-only baselines are necessary to interpret the behavioural role of episodic novelty. In the revision, we added episodic-only results to the GridWorld behavioural visualisations. Specifically, the revised heatmap figure now compares episodic-only, lifelong-only, and combined bonuses, and the exploration-diagnostics figure now includes episodic-only curves for exploration coverage, maximum within-episode revisit count, and cumulative unique states.
>
> These additions show that episodic-only exploration does reduce repeated within-episode revisits, but it lacks cross-episode memory and therefore often fails to systematically discover or stabilise later checkpoints in larger sequential layouts. This helps separate the two effects: episodic novelty is useful for preventing excessive revisitation, while the lifelong component is needed to bias exploration toward genuinely underexplored regions.
>
> ### 4. Explicit effect of revisit pressure $\beta$ and coordination complexity $L$
>
> We added an explicit comparison of final return as coordination complexity and revisit pressure vary. The new figure plots performance across GridWorld layouts that vary $L$ while keeping the task family fixed, and across layouts that vary geometric pressure while keeping $L$ fixed. This makes the empirical connection to the theory more direct: lifelong-only exploration degrades as $L$ or $\beta$ increases, whereas the combined bonus is substantially more stable.

---

> > ### Author Response · Authors · 2026-04-24
> >
> > ### 5. Cumulative unique states and Checkpoint-discovery-rate diagnostic
> >
> > We added cumulative unique-state coverage to the exploration diagnostics. This complements the per-episode coverage and revisit-count curves by measuring long-term discovery rather than only within-episode behaviour. The results show that the combined method initially discovers new states quickly and then plateaus once it finds a successful route, while episodic-only exploration tends to remain less directed and lifelong-only exploration often continues exploring without stabilising coordinated progress.
> >
> > We also added a checkpoint-discovery-rate diagnostic, measuring when agents first reach later checkpoints during training. This confirms that the combined bonus not only increases state coverage but also discovers task-relevant coordination states more reliably than episodic-only or lifelong-only exploration.
> >
> > ### 6. Equation 2 clarification
> >
> > We revised the text around Equation 2 to make the quantity more intuitive. In particular, $\tau_L$ is now described as the expected number of episodes required until all $L$ sequential checkpoints have been completed. This makes clear that $p_k$ measures the probability of advancing to the next checkpoint in a given episode, while $\tau_L$ measures overall task-completion efficiency.
> >
> > ### 7. Notation clarification: $\lambda_i$, $\mathcal{S}^i$, and $c_i$
> >
> > We clarified the notation in the main text. Here, $\lambda_i$ is an agent-specific exploration-mixing constant controlling how strongly the intrinsic bonus lower-bounds the probability that agent $i$ visits a local state; $\mathcal{S}^i$ is the local state space of agent $i$; and $c_i$ is an agent-specific constant controlling the scale of checkpoint revisitation in the polynomial revisit bound $N^i_k(s^{*}_{i,1}) \le c_i k^{\beta_i}$. We also clarified that $\beta_i$ captures the growth rate of this revisit count, and that the analysis uses $\beta=\max_i \beta_i$ as a conservative upper bound.
> >
> > ### 8. Figure 1(c): third coordination checkpoint
> >
> > We clarified the caption and surrounding text for Figure 1(c). The third checkpoint in that example corresponds to both agents arriving together in the target room after completing the earlier switch-door coordination events.
> >
> > ### 9. Appendix B.2 Step 5: tightness of the bound
> >
> > We agree that the expectation bound could be tightened by keeping the finite summation structure rather than upper-bounding by an infinite tail. We kept the looser form because it gives a clean sufficient condition for finite expected hitting time and preserves the dependence on the key quantity $\beta N/2$. We now state explicitly in the proof that the bound is not intended to be tight, and that sharper constants could be obtained by retaining the finite exponential summation. This does not change the qualitative conclusion: when $\beta N/2$ is large, the lifelong-only guarantee becomes weak, while the episodic/hybrid bounds remain independent of $\beta$ under the stated assumptions.
> >
> > ### 10. Minor corrections
> >
> > We corrected the reported minor textual issues and improved the relevant captions and notation explanations throughout the manuscript.

---

### Review · Reviewer_RoFu · 2026-04-11

**Summary Of Contributions:**

This paper identifies the failure mode "coordination de-synchronisation" in decentralised MARL, which depends on two independent factors: coordination complexity L (number of sequential checkpoints) and geometric revisit pressure β (how much the environment forces revisitation). A stylised theoretical framework shows that lifelong bonus guarantees decay polynomially controlled by βN/2, while a hybrid bonus inherits both a coordination floor from the episodic component and exploratory drive from the lifelong component. Theoretical bounds on expected hitting time are provided, and empirical evaluations on GridWorld, Overcooked, and StarCraft II support these claims.

Pros
1. The paper is well written and easy to read.
2. The problem is well defined, and the L × β decomposition is clean and provides actionable design guidance.
3. Well-constructed ablation study and thorough analysis. The GridWorld experiments control and test L and β independently, Overcooked and StarCraft II validate cooperative and combat tasks, respectively.

Cons
1. The hybrid scheme is not novel; the contribution of this work is the theoretical analysis, not the methodology. Badia et al. (ICLR 2020) first combined episodic and lifelong novelty multiplicatively in single-agent RL (Never Give Up), and this work should be cited. Additionally, Hernandez et al. (2025) directly extend NGU to multi-agent settings, which should also be discussed. The contribution should be reframed accordingly.
2. The proofs and theoretical guarantees rely on strong assumptions. For example, Assumption A6 (gated reachability) requires checkpoint locations to be unreachable until prior checkpoints are completed, which may not hold in Overcooked or StarCraft II. Assumptions A2 (exploration-mixing) and A3 (bounded cross-agent correlation) are also never experimentally validated, even in the tabular GridWorld setting where would be feasible.
3. The episodic coefficient inconsistency is concerning. The episodic coefficient is 0.005 for episodic-only but 1.0 for combined (Table 2), a 200× difference. The explanation that this keeps "effective magnitudes comparable" needs to be supported.

Adrià Puigdomènech Badia, Pablo Sprechmann, Alex Vitvitskyi, Daniel Guo, Bilal Piot, Steven Kapturowski, Olivier Tieleman, Martín Arjovsky, Alexander Pritzel, Andrew Bolt, and Charles Blundell. Never give up: Learning directed exploration strategies. In International Conference on Learning Representations, 2020.
Juan Hernandez, Diego Fernández, Manuel Cifuentes, Denis Parra, and Rodrigo Toro Icarte. Extending NGU to multi-agent RL: A preliminary study. In Latinx in AI Workshop at NeurIPS, 2025. arXiv:2512.01321.

**Audience:**

Yes

**Audience Explanation:**

This work addresses a gap in MARL exploration research with practical implications for intrinsic motivation design.

**Claims And Evidence:**

Yes

**Claims Explanation:**

The claims are well supported by systematic experiments in GridWorld, Overcooked, and StarCraft II. The experimental observations align with the theoretical analysis and predictions. There are only some minor gaps remaining that may need further investigation.

**Requested Changes:**

1. Add citation mentioned above and discuss.
2. Results are from 7 seeds with high variance. Significance test results should be provided to support the claim.
3. Why does the additive combination fail in GridWorld while the multiplicative combination works? Can the theory explain this?
4. In SmallCoor-NarrowCounter without communication (S=1), all hybrid methods score 0. Any explanation for this observation?
5. Lemma 4 and Assumption A6 require checkpoint locations to be unreachable until prior checkpoints are completed. Is this always true in Overcooked and StarCraft? If not, how does this affect the theoretical guarantees?

Minor:
P13 Figure 6 caption typo ("Figure 4").

---

> ### Author Response · Authors · 2026-04-24
>
> We thank the reviewer for the constructive comments. We revised the paper to clarify the contribution, strengthen the empirical support, and better delimit the scope of the theory.
>
> ### 1. Scope clarification and relation to NGU / multi-agent NGU
>
> We agree that the multiplicative combination of episodic and lifelong novelty is not itself a new exploration mechanism. We revised the framing of the paper accordingly and now explicitly cite NGU and the recent multi-agent NGU extension.
>
> The revised paper positions our contribution as theoretical and diagnostic rather than as the invention of a new hybrid bonus. NGU introduced the multiplicative combination of episodic and lifelong novelty in single-agent RL. The multi-agent NGU extension studies NGU-style exploration in MARL. Our paper addresses a different question: why lifelong novelty can induce coordination de-synchronisation in decentralised cooperative MARL, how this failure depends on coordination complexity $L$ and revisit pressure $\beta$, and why adding an episodic component mitigates this failure mode under the stated assumptions.
> We revised the introduction and related work to make this distinction explicit.
>
> ### 2. Assumption strength and applicability of the theoretical guarantees
>
> We agree that the assumptions are strong and should be interpreted carefully. We revised the manuscript to state that the lemmas provide sufficient-condition guarantees for a stylised class of sequential coordination problems, not universal guarantees for all MARL environments.
>
> In particular, A6 is exact for the controlled GridWorld tasks, where later checkpoints are structurally reachable only after earlier coordination events. In Overcooked and StarCraft II, the assumption should be understood more qualitatively: these domains contain bottlenecks, repeated handoff points, or repeated tactical configurations that create analogous revisit pressure, but they do not necessarily satisfy every formal assumption literally. We now state this distinction explicitly.
>
> For A2 and A3, we clarified that these are behavioural regularity assumptions connecting intrinsic reward to visitation probabilities and joint synchronisation probabilities. They are not guaranteed by the algorithm in arbitrary environments. We revised the theory section to explain when they are plausible and how they could be empirically checked in tabular settings. We also provided a diagnostics for A2 and A3 in a simple GridWorld and local-novelty only exploration setting in the appendix.
>
> ### 3. Episodic coefficient scaling
>
> We agree that the 200× coefficient difference needed stronger justification. The raw episodic-only and hybrid intrinsic rewards live on different numerical scales. Episodic-only novelty resets every episode and can remain persistently large, so a small coefficient is needed to avoid overwhelming the sparse extrinsic reward. In the hybrid case, the episodic signal is multiplied by the lifelong/RND term, which attenuates the realised intrinsic reward as training progresses.
>
> To support this explanation, we added a plot of coefficient-scaled realised intrinsic reward magnitudes during training in the appendix. The plot shows that, after scaling, episodic-only and hybrid intrinsic rewards are of comparable effective magnitude, despite using different raw coefficients. We also clarified this point in the hyperparameter section.
>
> ### 4. Statistical significance under high variance
>
> We added statistical significance tests to the main quantitative table. Because the final returns in sparse-reward coordination tasks are highly non-Gaussian and often discrete or bimodal, we use a non-parametric Mann–Whitney U test on final evaluation returns across seeds. We mark entries where the best-performing method significantly improves over the second-best method at (p<0.05).
>
> We also retained means and standard deviations over 7 seeds, since variance itself is informative in these sparse coordination tasks: small differences in early exploration can lead to qualitatively different success/failure outcomes. Given the small number of seeds, we present these tests as supportive rather than definitive.

---

> > ### Author Response · Authors · 2026-04-24
> >
> > ### 5. Why additive combination fails while multiplicative combination works
> >
> > We clarified this mechanism in the revised manuscript. In an additive bonus,
> >
> > $$ r_{\mathrm{add}}=\alpha r_{\mathrm{life}}+\eta r_{\mathrm{epi}}, $$
> >
> > the episodic component can dominate once the lifelong component decays, causing the method to behave similarly to episodic-only exploration. This preserves within-episode diversity but weakens the global discovery gradient toward states that are genuinely underexplored across training.
> >
> > In contrast, the multiplicative bonus acts as a gate: a state receives a high bonus only when it is both globally underexplored and episodically novel. This preserves the lifelong component’s directionality toward unexplored regions while using the episodic component to suppress repeated within-episode cycling. Our theory is therefore intended to support the multiplicative case; we do not claim that every additive weighting must fail, only that the simple additive variant we tested does not preserve the same interaction between temporal novelty scales.
> >
> > ### 6. Zero performance in SmallCoor-NarrowCounter without communication
> >
> > We clarified this result. SmallCoor-NarrowCounter without communication is an extreme sparse-coordination setting: agents must discover that the narrow counter is a useful handoff bottleneck while observing only local novelty. In terms of the theory, this corresponds to very small effective exploration and synchronisation constants, such as $\lambda_i$ and $\rho_{\min}$. The hybrid bonus does not solve the task in this regime because the agents rarely discover the relevant handoff convention in the first place.
> >
> > The result is therefore not a contradiction of the theory. The theory gives sufficient-condition guarantees when the exploration-mixing and synchronisation assumptions hold with non-negligible constants. In this layout without communication, those constants appear too small for reliable learning. When agents share novelty information through Summation or MACE, performance improves sharply, consistent with the interpretation that communication increases the effective constants while leaving the underlying revisit-pressure issue intact.
> >
> > ### 7. A6 in Overcooked and StarCraft II
> >
> > We revised the manuscript to clarify the scope of A6. The formal guarantees apply most directly to environments with explicit sequential gating, such as the GridWorld checkpoint tasks. Overcooked and StarCraft II are not claimed to satisfy A6 exactly. Instead, they are used to test whether the same qualitative mechanism appears in more complex domains where progress depends on repeated bottleneck interactions or repeated tactical configurations. We now state this distinction clearly in the theory discussion and limitations.
> >
> > ### 8. Minor correction
> >
> > We corrected the Figure 6 caption typo and checked the surrounding figure references.

---

### Decision · Action_Editor_eHwa · 2026-05-18

**Recommendation:** Accept with minor revision

**Additional Comments:**

First paragraph of 3.2: References to $\lambda_i$ and other not-yet defined constants should be replaced by some more generic description.

Assumption A2: The phrasing is a bit unclear here.  It may be clearer to write the proportionality relation you intend rather than give an English description.  Perhaps relatedly, as far as I can tell $\kappa_i$ is not actually defined (is it meant to be just the normalization constant?) and the size of the state space is mentioned but the relationship to the assumption is unclear.

Assumption A3: What does "is comparable to" mean here?  It would be clearer if you wrote down explicit inequalities relating the product and your constants to the joint probabilities.

Assumption A6: I more or less understand the goal here from the English description, but I am unclear on the intended precise meaning.  I think it is not quite correct as stated, because if the first $\ell-1$ checkpoints are completed badly, there may be no remaining time budget left.  Then there is the $H \geq L$ condition which seems irrelevant to the setup given and is a generic fact that will be true or not about the problem regardless of whether any checkpoints are completed.  Should this instead assume something like the existence of a joint policy that reaches all checkpoints within the horizon with probability 1 / sufficiently high probability / positive probability?

**Audience:**

Yes

**Audience Explanation:**

This paper addresses challenges in MARL which have substantial interest to members of the TMLR audience.

**Claims And Evidence:**

No

**Claims Explanation:**

The authors incorporated substantial feedback from the reviewers which lead to the paper being better positioned with accurate claims justified by the evidence.  Thus this criterion is very close to being satisfied, with one unresolved issue.  In particular, there has been significant improvement regarding the assumptions behind the analysis, but some of them are still not stated fully clearly and precisely, which is essential to fully understand the claims made in the paper.  Below I detail the three changes needed to the assumptions, plus one small copyediting issue.

---

> ### Author Response · Authors · 2026-06-12
>
> Thank you for the detailed guidance. We agree that the previous presentation of the assumptions was not sufficiently precise, and we have revised the relevant parts of Section 3.2, Section 3.4, Section 3.5, Appendix A, and Appendix B accordingly.
>
> First, in Section 3.2, we removed the early references to $\(\lambda_i\)$ and other constants before their formal definitions, replacing them with a generic statement that heterogeneous agents only change the behavioural and task-dependent constants introduced later in the assumptions.
>
> Second, for Assumption A2, we replaced the informal proportionality statement with an explicit normalised-bonus definition and two-sided inequalities. The normalisation term is now written explicitly, and we clarify that $\(\lambda_i\)$ and $\(\kappa_i\)$ are multiplicative comparison constants, not normalisation constants.
>
> Third, for Assumption A3, we removed the vague phrase “is comparable to” and now state explicit upper and lower inequalities relating the joint visitation probability to the product of the agents’ marginal visitation probabilities.
>
> Finally, for Assumption A6, we replaced the previous horizon-based $\(H\ge L\)$ formulation with an explicit staged feasibility assumption. The revised assumption defines stage budgets $\(h_1,\ldots,h_L\)$, stage events $\(E_{\ell,k}\)$, and constants $\(q_\ell\)$ that lower-bound the probability of reaching each checkpoint within its allocated stage, conditional on the earlier checkpoints having been reached. We also revised Lemma 4 and its proof to remove the old $\(H\ge L\)$ and $\(H-L+1\)$ arguments, and to state the hybrid guarantee as a finite-window bound, with the expected hitting-time statement
> only under the corresponding maintained lower-bound condition.
>
> These changes directly address the ambiguity in A2, A3, and A6, and make the assumptions mathematically explicit.